# Joint Localization and Activation Editing for Low-Resource Fine-Tuning

Wen Lai [1 2]   Alexander Fraser [1 2]   Ivan Titov [3 4]

## Abstract

Parameter-efficient fine-tuning (PEFT) methods, such as LoRA, are commonly used to adapt LLMs. However, the effectiveness of standard PEFT methods is limited in low-resource scenarios with only a few hundred examples. Recent advances in interpretability research have inspired the emergence of activation editing (or steering) techniques, which modify the activations of specific model components. Due to their extremely small parameter counts, these methods show promise for small datasets. However, their performance is highly dependent on identifying the correct modules to edit and often lacks stability across different datasets. In this paper, we propose Joint Localization and Activation Editing (JOLA), a method that jointly learns (1) which heads in the Transformer to edit (2) whether the intervention should be additive, multiplicative, or both and (3) the intervention parameters themselves - the vectors applied as additive offsets or multiplicative scalings to the head output. Through evaluations on three benchmarks spanning commonsense reasoning, natural language understanding, and natural language generation, we demonstrate that JOLA consistently outperforms existing methods.[1]

## 1. Introduction

Parameter-efficient fine-tuning (PEFT; Han et al., 2024) methods are widely used to adapt large language models (LLMs). However, popular PEFT approaches like LoRA (Hu et al., 2021) often struggle in low-resource settings with only a few hundred examples. Inspired by advances in interpretability research (Vig et al., 2020; Zhang & Nanda, 2023), *activation editing* techniques (Wu et al., 2024a; Yin et al., 2024) have emerged as an alternative. These methods modify model activations to adapt LLMs to new tasks, leveraging the intuition that LLMs encode many semantically meaningful properties in a coordinate-aligned (or even disentangled) manner. The activations can then be adjusted with simple operations such as pruning, rescaling or addition. Activation editing method avoid more complex transformations, such as the MLPs used in the original Adapters (Houlsby et al., 2019). Editable components in activation editing include bias terms (Ben Zaken et al., 2022), MLP layer outputs (Wu et al., 2024a), hidden states within MLP layers (Wu et al., 2024b), and attention head outputs (Yin et al., 2024).

Compared to standard PEFT methods like LoRA (Hu et al., 2021), activation editing modifies significantly fewer parameters. For example, in our experiments, the optimal LoRA configuration altered 0.826% of LLaMA-3-8B's (Dubey et al., 2024) parameters, whereas LoFIT (Yin et al., 2024) updated only 0.002%.[2] This drastic reduction makes activation editing particularly appealing for low-resource scenarios, where only a few hundred training examples are available.

However, activation editing's effectiveness is highly sensitive to the choice of modules. This selection is typically determined either by fixed hyperparameters – specifying which layers and component types to modify (Ben Zaken et al., 2022; Wu et al., 2024b) – or by additional methods that estimate the importance of different model components for a given task (Yin et al., 2024). Furthermore, existing approaches vary in their intervention strategies (e.g., additive vs. multiplicative modifications), with no clear consensus on which method is most effective across tasks and models. As a result, performance tends to be inconsistent across different datasets and models (see Table 1 and Figure 4).

To address these limitations, we propose **Joint Localization and Activation Editing** (**JOLA**), a method that, for a given task, jointly learns (1) which components to edit, (2) whether to apply additive, multiplicative, or combined interventions, and (3) the optimal intervention parameters – the vectors applied as additive offsets or multiplicative scalings

---

*Work done during Wen Lai's visit to The University of Edinburgh. [1]Technical University of Munich [2]Munich Center for Machine Learning [3]ILLC, University of Edinburgh [4]ILLC, University of Amsterdam. Correspondence to: Wen Lai <wen.lai@tum.de>, Ivan Titov <ititov@inf.ed.ac.uk>, Alexander Fraser <alexander.fraser@tum.de>.

*Proceedings of the 42nd International Conference on Machine Learning*, Vancouver, Canada. PMLR 267, 2025. Copyright 2025 by the author(s).

[1]The code for the method is released at https://github.com/wenlai-lavine/jola.

[2]Detailed comparisons are provided in Appendix A.

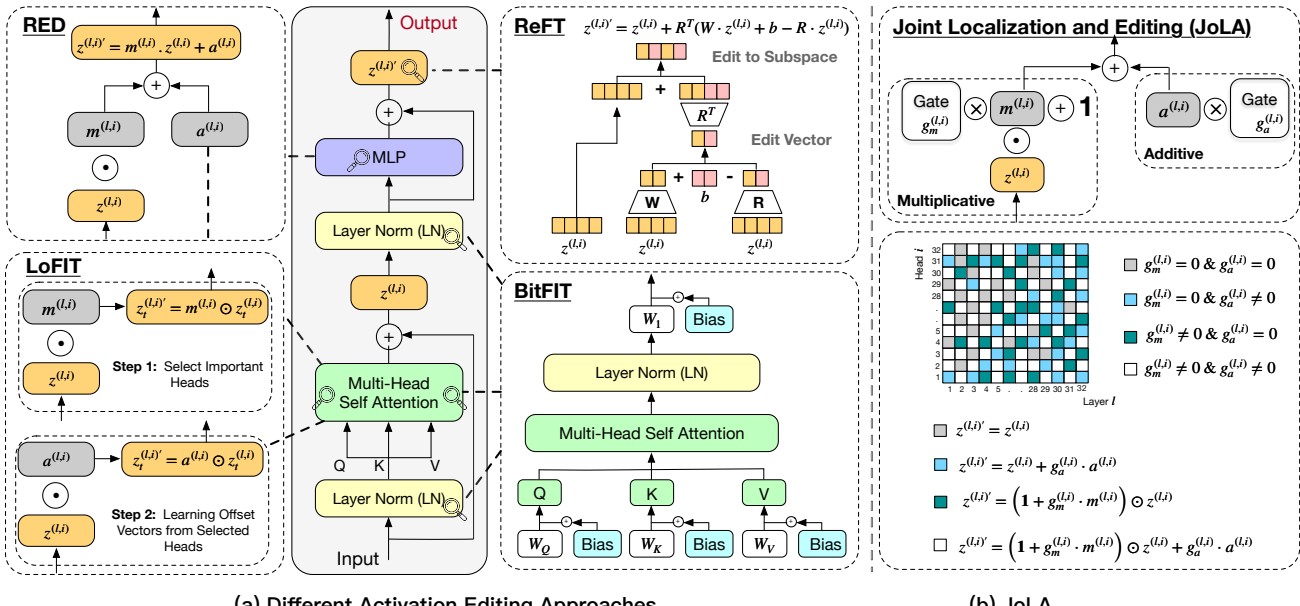

*Figure 1.* Comparison of previous representative activation editing methods with proposed JOLA. (a) includes BitFIT (Ben Zaken et al., 2022), which fine-tunes only the bias term; RED (Wu et al., 2024a) introduces scaling and bias vectors in the MLP layer; ReFT (Wu et al., 2024b), which fine-tunes the hidden layer representations; and LoFIT (Yin et al., 2024) intervenes with attention heads in two steps. (b) JOLA introduces a gating mechanism that dynamically selects and locates attention heads to modify the activation outputs. We compare activation changes ($z^{(l,i)'}$) across modules under two interventions (additive $a^{(l,i)}$ and multiplicative $m^{(l,i)}$), relative to the initial activation value ($z^{(l,i)}$).

to modules' outputs. Rather than relying on fixed heuristics or manual selection, JOLA dynamically identifies the most relevant components and applies targeted modifications to their activations.

To achieve this, JOLA uses HardConcrete gates with expected-L0 regularization, a technique previously employed for parameter (Louizos et al., 2018) and component pruning (Voita et al., 2019). This method encourages sparsity, ensuring that only a small subset of components is selected for editing, thereby reducing the number of interventions and, thus, the method's effective parameter count. We also observe that it appears sufficient to focus on heads' outputs rather than other component types, further reducing the parameter counts and enhancing the simplicity of the method. By combining additive offsets and multiplicative scalings, JOLA provides a flexible, data-efficient adaptation strategy.

We evaluate JOLA across three benchmark categories: commonsense reasoning, natural language understanding, and natural language generation. Experimental results on 26 tasks from the benchmarks (Hu et al., 2023; Wang et al., 2024b; Gehrmann et al., 2022) demonstrate that JOLA consistently outperforms existing methods in low-resource settings (as shown in Figure 4), delivering robust performance across various data scales and model sizes.

In summary, our contributions are as follows: (i) We introduce JOLA, a novel activation editing approach that jointly optimizes the selection of intervention components and the intervention strategy, specifically tailored for low-resource scenarios. (ii) We demonstrate that JOLA achieves stable and consistent performance across diverse tasks, addressing key limitations of existing methods. We further validate its effectiveness across different data scales and model sizes. (iii) We provide new insights into the role of attention heads in activation editing, showing that they are the most impactful components for fine-tuning.

## 2. Background

Activation editing in LLMs modifies intermediate activation outputs to steer model behavior. We categorize existing approaches into three types based on the transformation function applied to activations. Given an activation output $z_t^{(l,i)} \in \mathbb{R}^{d_l}$ for $i$-th component at layer $l$, the general transformation is:

$$z_t^{(l,i)'} = f(z_t^{(l,i)}), \qquad (1)$$

where $f(\cdot)$ determines the intervention type:

- **Additive methods** apply a bias vector $a_l^i \in \mathbb{R}^{d_l}$ : $z_t^{(l,i)'} = z_t^{(l,i)} + a^{(l,i)}$.

- **Multiplicative methods** scale activations as $z_t^{(l,i)'} = m^{(l,i)} \odot z_t^{(l,i)}$, where $m^{(l,i)} \in \mathbb{R}^{d_l}$ and $\odot$ is an element-wise product.
- **Hybrid methods** combine both transformations: $z_t^{(l,i)'} = m^{(l,i)} \odot z_t^{(l,i)} + a^{(l,i)}$.

Existing methods follow these paradigms but often rely on fixed selections of components for modification, limiting adaptability. For example, BitFit (Ben Zaken et al., 2022) updates bias terms, while RED (Wu et al., 2024a) employs per-dimension scaling vectors and bias vectors. ReFT (Wu et al., 2024b) applies fine-tuned low-rank hidden states with MLP layers, and LoFIT (Yin et al., 2024) intervenes in selected attention heads with additive bias vectors but requires manual selection. JOLA also modifies attention heads but unifies the processes of localization and intervention within a single framework, in contrast to LoFIT's rigid two-stage pipeline. A detailed comparative analysis between JOLA and LoFIT is provided in Appendix B.

## 3. Method

In this section, we introduce JOLA, a novel approach for fine-tuning LLMs in low-resource settings. We first identify two key challenges in existing approaches and present an analysis to better motivate our method (Section 3.1). We then propose a gated dynamic attention head selection mechanism to address these limitations (Section 3.2). Figure 1 illustrates the comparison of previous activation editing approaches and JOLA.

### 3.1. Motivation

Activation editing methods have demonstrated success in modifying Transformer components such as bias terms (Ben Zaken et al., 2022), MLP layers (Wu et al., 2024a), low-rank hidden state subspaces (Wu et al., 2024b), and specific attention heads (Yin et al., 2024). However, two critical questions remain underexplored: **Q1:** Which Transformer components are most crucial for effective activation editing? **Q2:** What combination of multiplicative and additive operations yields the best performance for intervention? Existing approaches predefine the components to edit and rely on fixed intervention strategies, such as simple multiplicative scaling, which limits adaptability and can lead to inconsistent performance across tasks, especially in low-resource scenarios. To address these questions, we conduct controlled experiments to compare the effectiveness of editing different Transformer components and analyze the relative contributions of multiplicative and additive operations.[3]

---

[3]Details on experimental setup and datasets are provided in Section 4.

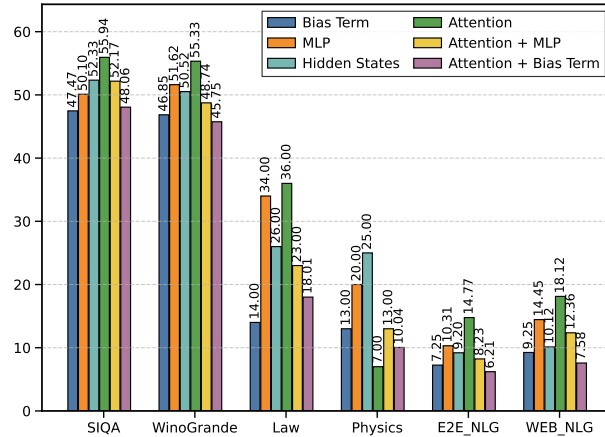

Figure 2. Performance comparison of activation editing across different Transformer modules: bias terms, MLP layers, hidden states, and attention heads.

**Q1: Component Selection.** We evaluate activation editing across four Transformer components: bias terms, MLP layers, hidden states, and attention heads[4]. Figure 2 shows that attention heads are the most impactful component to target. Unlike other modules, which primarily refine intermediate representations, attention heads encode key semantic relationships and are critical for reasoning and task adaptation (Ren et al., 2024). Interestingly, combining interventions across multiple components tends to degrade performance, which we interpret as a form of overfitting. These findings highlight the importance of carefully selecting where we intervene (i.e., the choice and location of components) and having fewer interventions. Further discussion is provided in Appendix C.

**Q2: Scaling vs. Offset Operations.** Activation editing typically involves two operations: scaling (multiplicative) and offset (additive) adjustments. To evaluate their relative importance, we conduct ablation studies isolating each operation. As shown in Figure 3, bias offsets consistently contribute more to performance improvements than scaling. We hypothesize that this behavior arises because the bias directly adjusts the latent representations, facilitating fine-grained task-specific adaptation while retaining the features of the pre-trained model. In contrast, scaling modifies the activations uniformly, which may introduce unintended distortions. These findings motivate our approach: JOLA incorporates both operations but prioritizes offset interven-

---

[4]Intervention on "hidden states" follows ReFT, which applies learned modifications directly to the output of the MLP sub-layer—i.e., after the nonlinearity—within the transformer block. In contrast, intervention on "bias terms" follows BitFit, which fine-tunes only the existing bias parameters in the model, such as those in linear projections and layer normalization. Notably, BitFit modifies only the biases in the linear layers surrounding these mechanisms and does not introduce new parameters.

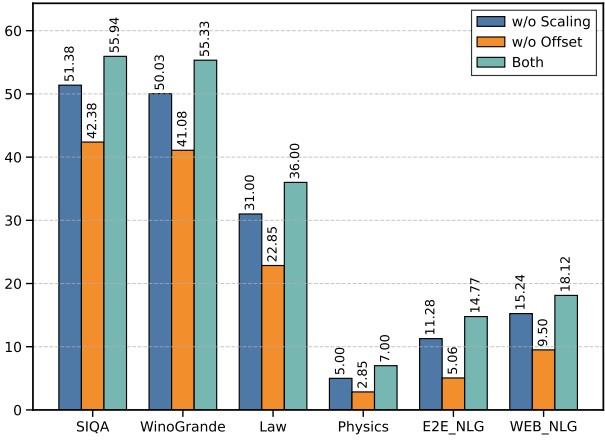

*Figure 3.* Comparison of the performance impact of scaling factors versus bias offsets in activation editing.

tions for more effective adaptation.

### 3.2. Joint Localization and Editing

Based on our insights from Section 3.1, JOLA focuses on adaptive attention head interventions to maximize activation editing effectiveness. Existing methods like LoFIT (Yin et al., 2024) require manual hyperparameter tuning to select the number of attention heads and cannot adjust the chosen heads during training. Moreover, their head selection criteria do not necessarily align with interventions. For instance, LoFIT employs multiplicative variables to determine head selection before restarting training with additive-only interventions. To address these limitations, we propose a method that jointly learns which heads to modify while optimizing intervention parameters (i.e., vectors $m^{(l,i)}$ and $a^{(l,i)}$).

We extend the hybrid intervention method from Section 2 by introducing two scalar gates, $g_a^{(l,i)}$ and $g_m^{(l,i)}$, both in $[0,1]$. This results in the transformation:

$$z_t^{(l,i)'} = (\mathbf{1} + g_m^{(l,i)} \cdot m^{(l,i)}) \odot z_t^{(l,i)} + g_a^{(l,i)} \cdot a^{(l,i)}, \quad (2)$$

where $\mathbf{1} \in \mathbb{R}^{d_l}$ is a vector of ones. The transformation is designed so that when both gates are closed ($g_a^{(l,i)} = g_m^{(l,i)} = 0$), it reduces to the identity map, effectively disabling the intervention for that head. By using separate gates, the model can learn to apply additive and multiplicative modifications independently.

Since our goal is to apply activation editing to a small, adaptively chosen subset of heads, we encourage the gates to be exactly zero where intervention is unnecessary. To achieve this, we use expected-$L_0$ regularization, a technique originally introduced by Louizos et al. (2018) for pruning neural network weights. This approach has since been successfully applied to tasks such as head pruning (Voita et al.,

2019) and extracting reasoning paths in graph neural networks (Schlichtkrull et al., 2021).

During training, each gate is modeled as a scalar stochastic variable drawn from a Hard-Concrete distribution (Louizos et al., 2018),

$$g_a^{(l,i)} \sim P(g_a^{(l,i)} \mid \phi_a^{(l,i)}), \quad g_m^{(l,i)} \sim P(g_m^{(l,i)} \mid \phi_m^{(l,i)}). \quad (3)$$

To clarify, these gates do not take any input – each gate is simply an instance of the Hard-Concrete distribution with a single learnable parameter.

The Hard-Concrete distribution is a mixed discrete-continuous distribution over $[0,1]$, with point masses at $0$ and $1$ and a continuous density over $(0,1)$. The closed-form probability of a gate being non-zero (e.g., $1 - P(g_a^{(l,i)} = 0 \mid \phi_a^{(l,i)})$), is used to define a sparsity-inducing regularizer:

$$L_C(\phi) = \sum_{l,i} \Big(1 - P(g_a^{(l,i)} = 0 \mid \phi_a^{(l,i)}) \\ + 1 - P(g_m^{(l,i)} = 0 \mid \phi_m^{(l,i)})\Big). \quad (4)$$

The overall training objective balances task-specific performance with sparsity:

$$L(\mathbf{m}, \mathbf{a}, \phi) = L_{xent}(\mathbf{m}, \mathbf{a}) + \lambda L_C(\phi), \quad (5)$$

where $L_{xent}$ is the standard cross-entropy loss, and $\lambda$ controls the trade-off between performance and sparsity. Optimization is performed over all intervention parameters: $\phi$, $\mathbf{m}$ and $\mathbf{a}$.

As with parameter pruning (Louizos et al., 2018), the expected value of each gate can be computed. The interventions with very low expected gate value (i.e., $\mathbb{E}[g_m^{(l,i)}] < \epsilon$) can be disregarded with no effect on JOLA performance. For the remaining heads, the gate is set during inference to its expected value, $\mathbb{E}[g_a^{(l,i)}]$ and $\mathbb{E}[g_m^{(l,i)}]$, removing any randomness and ensuring stability in the inference stage.

By dynamically selecting and adapting attention head interventions, JOLA achieves efficient and effective activation editing, overcoming the limitations of previous methods. Our approach ensures robust, data-efficient adaptation across diverse tasks, making it well-suited for low-resource settings.

## 4. Experiments

### 4.1. Datasets and Tasks

We evaluate JOLA on three diverse tasks: commonsense reasoning, natural language understanding, and natural language generation. Additional details regarding the datasets can be found in Appendix D.

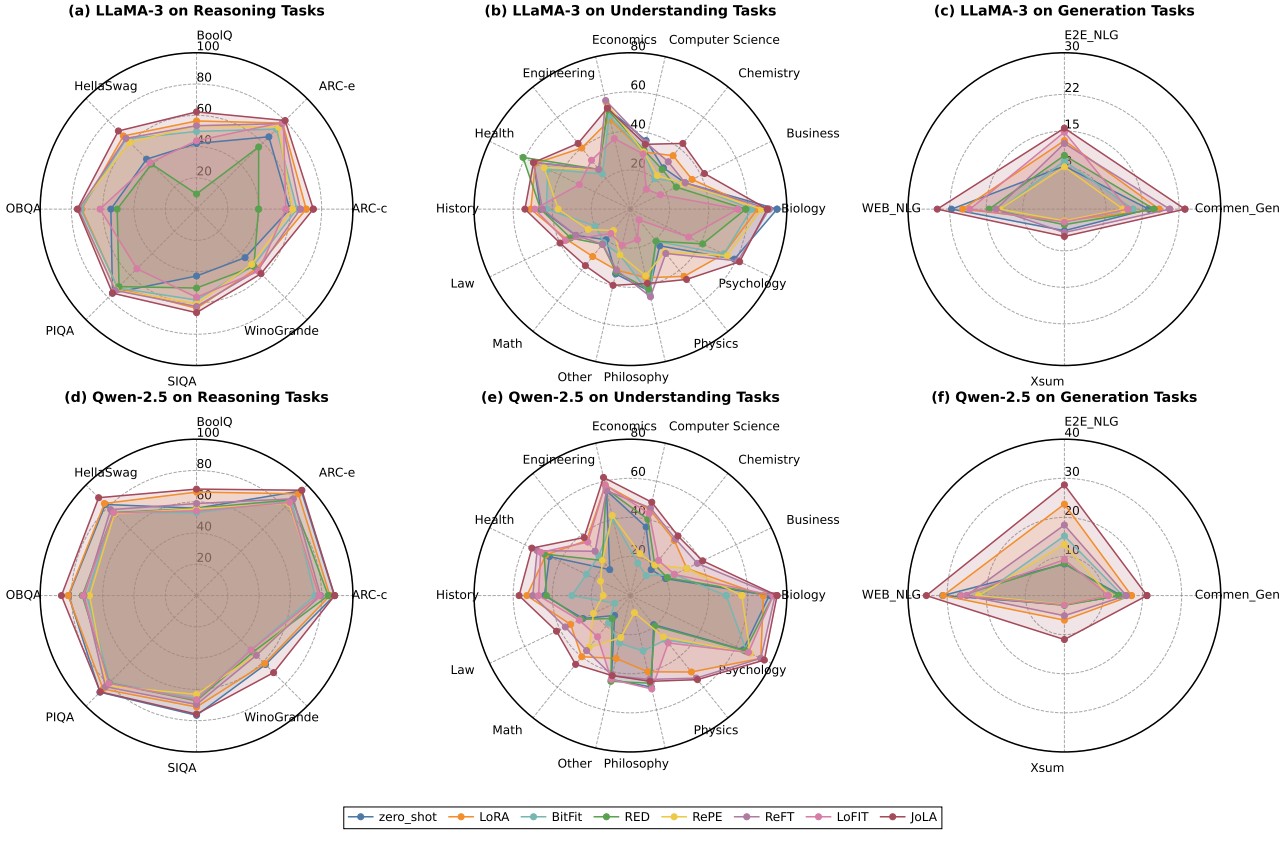

*Figure 4.* Performance comparison of JOLA and baseline methods across commonsense reasoning, natural language understanding, and natural language generation tasks for LLaMA-3 (Dubey et al., 2024) and Qwen-2.5 (Yang et al., 2024).

**Commonsense Reasoning.** For commonsense reasoning, we utilize a widely adopted benchmark (Hu et al., 2023; Wu et al., 2024b) containing 8 datasets: ARC-c and ARC-e (Clark et al., 2018), BoolQ (Clark et al., 2019), HellaSwag (Zellers et al., 2019), OBQA (Mihaylov et al., 2018), PIQA (Bisk et al., 2020), SIQA (Sap et al., 2019), and Wino-Grande (Sakaguchi et al., 2021). These datasets consist of multiple-choice questions, where the model must directly generate the correct option without providing explanations.

**Natural Language Understanding.** We evaluate on the MMLU-Pro benchmark (Wang et al., 2024b), covering 14 domains: Biology, Business, Chemistry, Computer Science, Economics, Engineering, Health, History, Law, Math, Philosophy, Physics, Psychology, and Others. Each task requires selecting the correct answer from ten options, testing the model's broad knowledge and reasoning capabilities.

**Natural Language Generation.** For generation tasks, we select 4 datasets from GEM benchmark (Gehrmann et al., 2022), including CommonGen (Lin et al., 2020) for concept-to-sentence generation, E2E (Novikova et al., 2017) and WebNLG (Gardent et al., 2017) for data-to-text generation,

and XSum (Narayan et al., 2018) for abstractive summarization of long documents. This selection ensures a diverse evaluation of generation tasks, including coherence, informativeness, and abstraction.

### 4.2. Baselines

We compare JOLA against a range of state-of-the-art baselines: (1) **Zero-Shot**: Direct evaluation of pre-trained large language models (LLMs) without fine-tuning, including LLaMA-3 (Dubey et al., 2024) and Qwen-2.5 (Yang et al., 2024). (2) **Parameter-Efficient Fine-Tuning**: LoRA (Hu et al., 2021), a method for efficient fine-tuning by injecting trainable low-rank updates into the model's weights. (3) **Activation editing during training**: BitFiT (Ben Zaken et al., 2022), a method that fine-tunes only the bias terms of the model; RED (Wu et al., 2024a), which adds scaling and bias vectors to the outputs of MLP layer; ReFT (Wu et al., 2024b), which directly intervenes on task-specific hidden states with MLP Layers, and LoFIT (Yin et al., 2024), a two-stage method that selects task-relevant attention heads and applies bias tuning. (4) **Activation editing during inference**: RePE (Zou et al., 2023), which modifies repre-

*Table 1.* **Main Results:** Average performance comparison (Accuracy, BLEU, ROUGE-L, BERTScore) of different activation editing methods across reasoning, understanding, and generation tasks for LLaMA-3.1 and Qwen-2.5 models. The best results in each category are highlighted in bold.

| | Llama-3.1-8B-Instruct | | | | | Qwen2.5-7B-Instruct | | | | |
| | Reasoning | Understanding | Generation | | | Reasoning | Understanding | Generation | | |
| | ACC (↑) | ACC (↑) | BLEU (↑) | ROUGE-L (↑) | BERTScore (↑) | ACC (↑) | ACC (↑) | BLEU (↑) | ROUGE-L (↑) | BERTScore (↑) |
|---|---|---|---|---|---|---|---|---|---|---|
| zero_shot | 53.70 | 40.00 | 12.56 | 36.70 | 77.23 | 78.65 | 37.21 | 14.03 | 34.29 | 78.52 |
| LoRA | 66.58 | 42.07 | 13.27 | 36.97 | 77.74 | 78.28 | 46.22 | 19.46 | 45.34 | 82.40 |
| BitFit | 63.05 | 35.02 | 9.25 | 28.81 | 74.83 | 69.25 | 28.72 | 13.47 | 33.10 | 77.89 |
| RED | 46.19 | 37.33 | 11.24 | 32.40 | 76.24 | 71.52 | 38.76 | 12.81 | 34.75 | 77.52 |
| RePE | 63.61 | 35.54 | 8.49 | 27.61 | 74.30 | 69.85 | 29.15 | 12.19 | 33.07 | 76.98 |
| ReFT | 65.95 | 40.89 | 12.60 | 36.89 | 77.21 | 72.69 | 47.74 | 16.02 | 37.40 | 79.74 |
| LoFIT | 56.19 | 27.76 | 11.88 | 32.09 | 76.71 | 69.93 | 43.13 | 12.31 | 34.68 | 77.16 |
| JOLA | **70.55** | **47.00** | **17.07** | **40.65** | **80.54** | **82.40** | **51.57** | **24.00** | **50.23** | **85.90** |

sentations derived from contrastive prompts.

## 4.3. Implementation

We conduct experiments using the *Llama-3.1-8B-Instruct* (8B) and *Qwen2.5-7B-Instruct* (7B) models as the primary base models. Both are publicly available via the Huggingface repository[5]. To study the impact of model size, we also experiment with smaller (*Llama-3.2-1B-Instruct*, *Llama-3.2-3B-Instruct*) and larger (*Llama-3.1-70B-Instruct*) model variants. For all datasets, we sample 200 examples to simulate low-resource scenarios, with further analysis of data size effects provided in Section 6. The prompt templates used in our method are also included in the Appendix E. The Hard-Concrete distribution has two associated scalar parameters: a scale parameter and a temperature parameter. Following prior work on sparsification (Voita et al., 2019; Louizos et al., 2018), we train only the scale parameter and fix the temperature to 0.33. In all baseline experiments, we observe that the choice of hyperparameters significantly affected performance across different tasks. To address this, we conduct a hyperparameter search for each method, selecting five hyperparameters and averaging the results. The final outcomes are presented in Table 1 and Figure 4. More details on the training setup, computational resources, and hyperparameter selection process are provided in Appendix F.

## 4.4. Evaluation Metrics

We employ exact match accuracy as the evaluation metric for commonsense reasoning and natural language understanding tasks. For natural language generation, we use BLEU (Papineni et al., 2002), ROUGE-L (Lin, 2004) and BERTScore (Zhang et al., 2019) scores as implemented in the GEM benchmark (Gehrmann et al., 2022).

---
[5] https://huggingface.co

## 5. Results

This section evaluates the performance of our proposed method, JOLA, in comparison with various baselines. Table 1 presents the average performance for all methods across the three tasks, while Figure 4 illustrates the results for individual subtasks. More detailed numerical results can be found in Appendix G.

**Performance of Activation-Based Baselines.** Activation editing baselines exhibit varying levels of success across tasks, but their sensitivity to hyperparameter selection and layer intervention limits their consistency. For example, BitFit (Ben Zaken et al., 2022) is quite sensitive to the placement of bias terms within the model. Adjusting bias terms in dropout layers or attention mechanisms results in performance fluctuations, particularly in low-data scenarios. Similarly, RED (Wu et al., 2024a) depends on the specific positions where scaling and bias vectors are introduced, leading to inconsistent results. RePE (Zou et al., 2023) is highly sensitive to the quality of activation representations across tasks, making it challenging to generalize its performance. ReFT (Wu et al., 2024b) achieves moderate success by intervening on selected layers but faces challenges in determining the optimal number and choice of layers. LoFIT (Yin et al., 2024), while effective in leveraging task-relevant attention heads, struggles to maintain consistency across tasks.

**Performance of LoRA.** LoRA achieves noticeable improvements over zero-shot baselines and, somewhat surprisingly, outperforms previous activation editing methods across all tasks when its rank hyperparameter is appropriately tuned. In tasks such as natural language generation, LoRA achieves higher BLEU and ROUGE-L scores, highlighting its ability to generate coherent outputs.

**Performance of JOLA.** Our proposed method, JOLA, consistently outperforms all baselines across the three tasks by a significant margin. This can be attributed to JOLA's dynamic gated selection mechanism. Unlike LoFIT (Yin et al.,

2024), which requires manual selection of attention heads, JOLA's mechanism enables the modifications to less relevant heads to gradually "die off" during training reducing to the heads of the base model, improving robustness and adaptability. In commonsense reasoning, JOLA achieves an average improvement of 3.97% over the best-performing baseline (LoRA) in LLaMA-3, as shown in Table 1. For natural language understanding, JOLA demonstrates consistent performance across diverse domains in the MMLU-Pro benchmark (Wang et al., 2024b) across all 14 subtasks as illustrated in Figure 4. In natural language generation tasks, JOLA achieves higher BLEU, ROUGE-L and BERTScore scores compared to activation-based baselines and LoRA.

# 6. Analysis

In this section, we present a detailed analysis of JOLA through ablation studies (Section 6.1, Section 6.2, and Section 6.3), an exploration of gate status during training (Section 6.4), and evaluations across varying data and model sizes (Section 6.5). Unless otherwise specified, the analyses in this section are conducted on selected tasks, including *SIQA*, *WinoGrande*, *Law*, *Physics*, *E2E_NLG*, and *WEB_NLG*. In addition, we provide a case study to better visualize the advantages of JOLA in Appendix H.

| | Reasoning | | Understanding | | Generation | |
|---|---|---|---|---|---|---|
| | SIQA | WinoGrande | Law | Physics | E2E_NLG | WEB_NLG |
| MLP w/o gate | 50.10 | 51.62 | 34.00 | 20.00 | 10.31 | 14.45 |
| MLP with gate | **52.46** | **52.43** | **36.00** | **23.00** | **11.23** | **16.25** |
| Attention w/o gate | 55.94 | 55.33 | 36.00 | 7.00 | 14.77 | 18.12 |
| Attention with gate | **66.22** | **58.33** | **40.00** | **46.00** | **15.54** | **24.39** |
| Attention + MLP w/o gate | 52.17 | 48.74 | 23.00 | 13.00 | 8.23 | 12.36 |
| Attention + MLP with gate | **53.28** | **52.07** | **27.00** | **16.00** | **10.42** | **14.83** |

*Table 2.* **Ablation 1:** Impact of MLP and Attention interventions with/without gate mechnism on model performance across tasks.

## 6.1. Ablation 1: Gate Mechanism

Dynamic gating attention head selection is central to the performance of JOLA, as detailed in Section 3.2. To evaluate its necessity, we compare models with and without the gating mechanism. As illustrated in Table 2, the gating mechanism substantially improves task performance, both when intervening in attention heads and MLP layers. We speculate that this improvement arises because certain attention heads can be modified more effectively to achieve the desired behavior in a generalizable way, whereas modifying others may disrupt the model. The gating mechanism can selectively adjust the activation outputs of relevant attention heads, avoiding excessive or unnecessary edits that could harm performance. In contrast, models without this gating mechanism fail to differentiate between "editable" and less "editable" heads, resulting in performance instability.

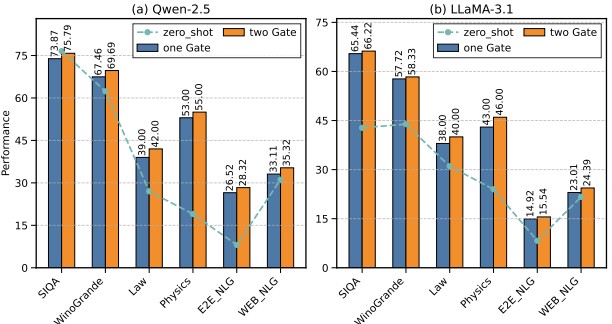

*Figure 5.* **Ablation 2:** Performance comparison of models with separate gating units for scaling and offset vectors versus a shared gating unit.

## 6.2. Ablation 2: Number of Gates

In Equation (2), we employ separate gating units for the scaling vector and the bias vector. To investigate the impact of this design, we compare configurations where each vector has its own gate with configurations where both vectors share a single gate. In the latter case, the heads, if selected, is always updated both in the additive and multiplicative fashion. As illustrated in Figure 5, although the shared gating configuration achieves a performance improvement over the zero-shot baseline, it underperforms compared to the configuration with separate gates. This suggests that the choice of intervention should depend on what role the head plays in a given task. Using independent gating units enables fine-grained control over each vector's contribution, facilitating more precise task-specific adjustments and preventing over-modification of the activation outputs.

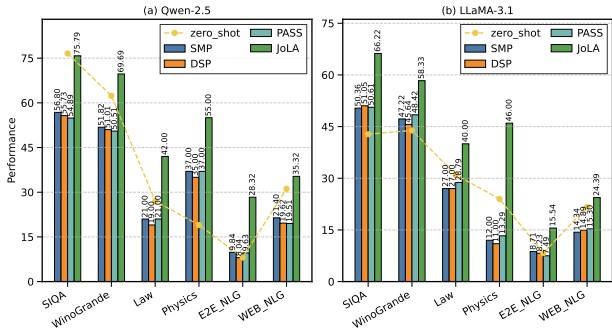

*Figure 6.* **Ablation 3:** Comparison of different head selection strategies: SMP, DSP, PASS, and JOLA.

## 6.3. Ablation 3: Different Head Selection Strategies

Head selection is a critical component of JOLA's design. To evaluate whether alternative selection strategies could achieve similar outcomes, we compare JOLA with three established methods: (1) **SMP** (Zhang et al., 2021), which trains a separate pruner to rank and identify attention heads

that are less important for the task; (2) **DSP** (Li et al., 2021), which employs Gumbel-Softmax (Jang et al., 2017) to iteratively select the top-K heads; and (3) **PASS** (Ding et al., 2024), which uses robust optimization to enforce deterministic sparsity.

As shown in Figure 6, JOLA outperforms these methods, especially in low-resource scenarios. SMP's reliance on large datasets for training the pruner makes it ill-suited for sparse data. DSP's iterative selection process is highly sensitive to noisy gradients from small datasets, leading to unstable or incorrect selection decisions. While PASS achieves deterministic sparsity, its regularization objective overfits to limited data distributions, resulting in suboptimal gate decisions. By contrast, JOLA's stochastic gating mechanism effectively balances exploration and exploitation, allowing it to adaptively identify important heads even in low-data settings.

### 6.4. Gate Status during Traning

To further investigate the behavior of the dynamic gating mechanism, we analyzed the probability of the multiplicative gate ($g_m$) and additive gate ($g_a$) being "closed" (i.e., set to 0) during training on the OBQA dataset (Mihaylov et al., 2018). As shown in Figure 7, both gates are initially "open" at the beginning of training (batch 1), allowing all attention heads to be editable. As training progresses, the probability of gates being "closed" increases, in that way the approach decides that these heads do not need to be modified. Interestingly, the multiplicative gate is more frequently "turned off" in the later stages of training. This observation supports our conclusion in Section 3.1 (**Q2**) that the additive gate $g_a$ has a greater impact on final model performance. To reiterate, we do not deactivate any attention heads. Instead, we selectively determine where to apply interventions. When both $g_a$ and $g_m$ are set to 0, the computation for that head remains identical to the original model, effectively bypassing intervention. By the end of training on OBQA, 86% of heads have $g_a = 0$, and 94% have $g_m = 0$, reflecting strong sparsity in applied edits.

### 6.5. Further Analysis

**Different Data Size**   To evaluate JOLA's robustness across different data scales, we conduct experiments using both small (100–1,000) and large (1,000–100,000) training examples sampled from the SIQA (Sap et al., 2019) and WinoGrande (Sakaguchi et al., 2021) datasets. As shown in Figure 8, JOLA consistently outperforms all baselines—even with as few as 100 training examples—demonstrating strong effectiveness in extreme low-resource settings. Performance improves steadily as the number of training examples increases from 100 to 10,000, highlighting JOLA's adaptability to varying data availabil-

ity. At intermediate scales (5,000–10,000 samples), JOLA remains competitive with or slightly outperforms strong parameter-efficient fine-tuning methods such as LoRA. When scaling further to 20,000 and 100,000 examples, JOLA shows a modest performance gap relative to LoRA, which is expected given that JOLA updates significantly fewer parameters. Nonetheless, it continues to be on par with LoRA's performance. These results demonstrate that JOLA not only effective in low-resource scenarios but also scales effectively to large datasets, making it a practical solution for both data-scarce and data-rich real-world applications.

**Different Model Size**   To evaluate the scalability of JOLA with respect to model size, we test three variants: *Llama-3.2-1B-Instruct*, *Llama-3.2-3B-Instruct*, and *Llama-3.1-70B-Instruct*. As shown in Figure 9, JOLA consistently delivers significant performance improvements across all model sizes. Notably, larger models benefit more substantially from JOLA's dynamic selection mechanism, as they inherently possess greater redundancy in attention heads. This finding highlights JOLA's scalability and effectiveness in optimizing large-scale models while maintaining robust performance in low-data scenarios.

## 7. Related Work

**Low-Resource Fine-tuning.**   Recent advancements in LLMs have transformed a wide range of NLP tasks (Zhao et al., 2023). However, efficiently adapting these models to diverse applications remains challenging, especially in low-resource settings. Parameter-efficient fine-tuning (PEFT) methods (Hu et al., 2021; Dettmers et al., 2024), which update a small subset of parameters or integrate new modules (Houlsby et al., 2019), achieve performance comparable to full fine-tuning across various tasks (Wang et al., 2024a). Yet, mitigating overfitting in low-resource scenarios remains a key challenge. Activation editing techniques (Ben Zaken et al., 2022; Wu et al., 2024a;b; Yin et al., 2024) offer a lightweight approach to model adaptation, often argued to be more data-efficient than standard PEFT methods.

**Pruning.** Neural network pruning (Cheng et al., 2024) aims to reduce model complexity and computational demands by removing less important or redundant components. Our approach builds on pruning techniques, specifically expected-$L_0$ regularization (Louizos et al., 2018). However, rather than pruning heads, our goal is to modify a selected subset of heads while keeping the rest intact. Subnetwork pruning techniques (e.g., Frantar & Alistarh, 2023; Sun et al., 2023) seek to identify an effective subnetwork, often tailored to a specific task, domain, or language. However, their primary objective is typically to match the performance of the full model rather than to specialize it for a particular task.

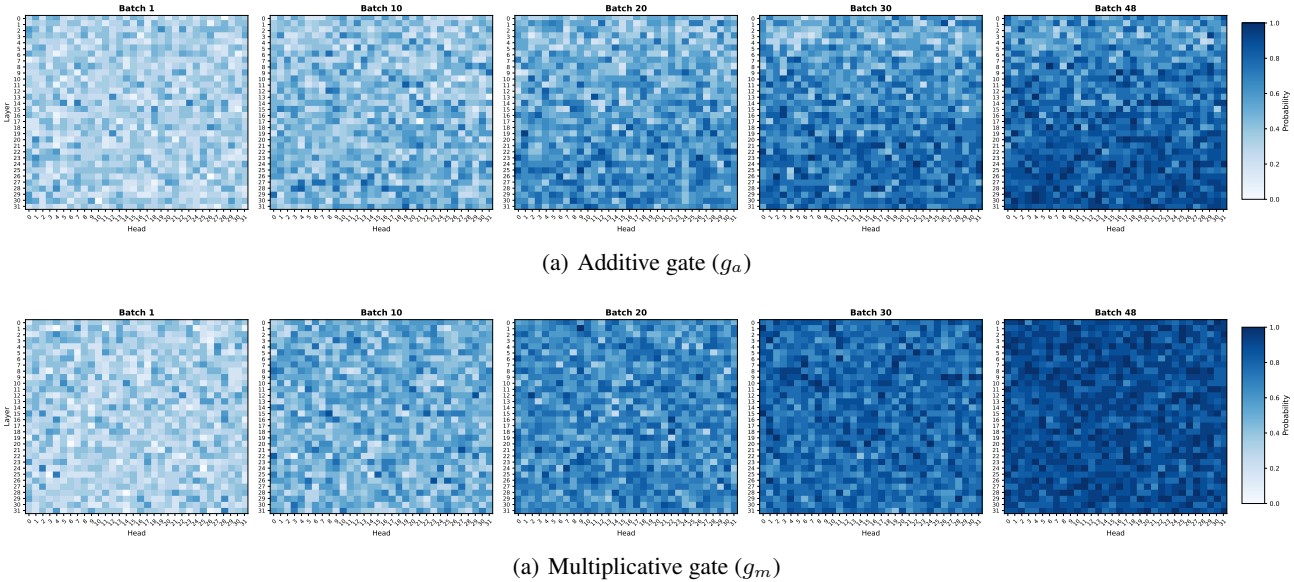

(a) Additive gate ($g_a$)

(a) Multiplicative gate ($g_m$)

*Figure 7.* Gate pruning probabilities for the additive gate ($g_a$) and multiplicative gate ($g_m$) during training on the OBQA dataset. A probability of 1 indicates a fully closed gate for the corresponding attention head.

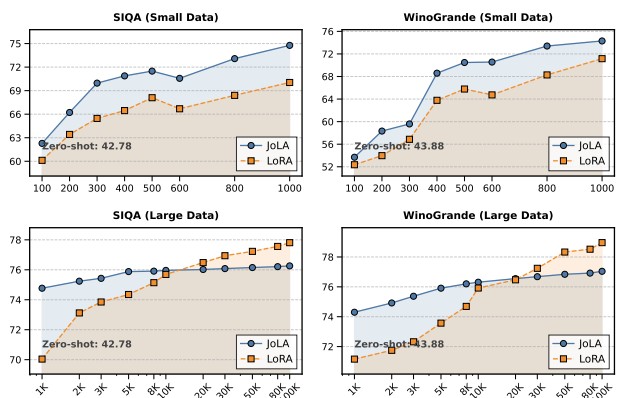

*Figure 8.* Performance of JOLA across different data sizes, evaluated on the SIQA and WinoGrande datasets.

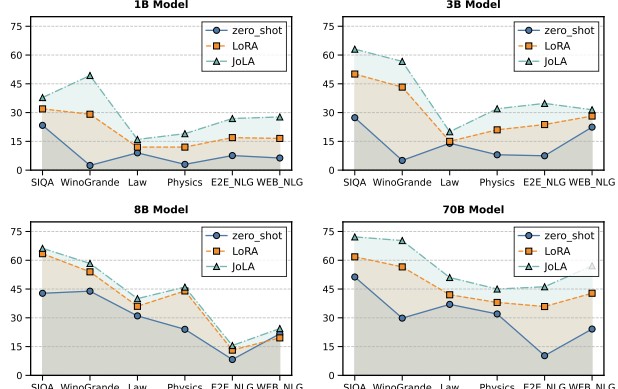

*Figure 9.* Performance comparison of JOLA across different model sizes: Llama-3.2-1B-Instruct, Llama-3.2-3B-Instruct, and Llama-3.1-70B-Instruct.

**Sparse Fine-tuning.** Sparse finetuning (Dao et al., 2022; Thangarasa et al., 2023) is a technique for adapting LLMs to specific tasks or datasets while only updating a small subset of the model's parameters. Our approach shares similarities with sparse fine-tuning, a technique commonly used in multilingual modeling (Nooralahzadeh & Sennrich, 2023; Choenni et al., 2024), where languages are typically assumed to be encoded modularly. Sparse fine-tuning identifies specific components (e.g., heads), fine-tunes all their parameters, and discards the others. In contrast, JOLA adjusts the activations of selected components while keeping the rest intact. While the goal of sparse fine-tuning is often to match the performance of the full model using a smaller version, our aim is not only to reduce model size but to enhance performance over the full model.

# 8. Conclusions

In this paper, we introduce JOLA, a novel approach to low-resource fine-tuning that jointly learns to dynamically localize the attention heads for targeted intervention and determine effective editing strategies using multiplicative scaling and/or additive bias vectors. We observe that attention heads are more effective than other model components in activation editing, offering a novel perspective for future research. Extensive experiments and ablation studies demonstrate the robustness of our method in low-data settings and across model scales, highlighting the importance of joint component selection and activation editing.

## Acknowledgments

The work is partially supported by the Dutch National Science Foundation (NWO Vici VI.C.212.053). The authors thank Rochelle Choenni for the suggestions. The work was supported by the European Research Council (ERC) under the European Union's Horizon Europe research and innovation programme (grant agreement No. 101113091) and by the German Research Foundation (DFG; grant FR 2829/7-1).

## Impact Statement

This paper presents work whose goal is to advance the field of Machine Learning. There are many potential societal consequences of our work, none of which we feel must be specifically highlighted here.

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

# A. Comparison with Full Parameter Fine-Tuning, PEFT and Traditional Activation Editing

To better demonstrate the advantages of our approach over full-parameter fine-tuning, LoRA (Hu et al., 2021), and existing activation editing methods, we compare them across five key dimensions: the percentage of modified parameters (both trainable and active), intervention modules, dynamic localization of interventions, data efficiency, and robustness across diverse tasks. Using fine-tuning of LLaMA-3-8B (Dubey et al., 2024) as a representative case, we summarize the differences in Table 3.

**Number of Parameters.** We distinguish between *trainable* parameters—those updated during training—and *active* parameters—those used during inference. *Full fine-tuning* updates and utilizes all model parameters, resulting in 100% trainable and active parameters. *LoRA* introduces low-rank adapters into each weight matrix. For LLaMA-3-8B with rank $r = 8$, this corresponds to approximately 0.2605% of the parameters being both trainable and active (Hu et al., 2021).

For activation editing, we take LoFIT and JOLA as examples. In JOLA, trainable parameters include: (1) Multiplicative scaling vectors $m^{(l,i)}$ and additive bias vectors $a^{(l,i)}$ for each attention head; (2) HardConcrete gate parameters $\phi_m^{(l,i)}$ and $\phi_a^{(l,i)}$ that determine head selection. All of these are optimized during training. However, due to $L_0$ regularization, most gates are pushed toward zero, effectively pruning the majority of heads. At inference, only the heads with non-zero expected gate values remain active, and only their associated $m^{(l,i)}$ and $a^{(l,i)}$ are applied. In contrast, LoFIT pre-selects a fixed subset of attention heads in a two-stage process: (1) Training all heads, then (2) Fine-tuning only a selected subset.

The number of trainable parameters can be approximated as:

$$P_{\text{trainable}} = \frac{D_{\text{attn}} \times (N_{\text{multi}} + N_{\text{add}} + N_{\text{gate}})}{P_{\text{LLMs}}} \tag{6}$$

where $D_{\text{attn}}$ is the dimension of each attention head, $N_{\text{multi}}$, $N_{\text{add}}$, and $N_{\text{gate}}$ are the numbers of multiplicative, additive, and gating parameters, respectively, and $P_{\text{LLMs}}$ is the total number of parameters in the base LLM.

JOLA and LoFIT exhibit similar numbers of active parameters at inference. Minor variations across tasks are expected: JOLA selects heads dynamically based on the input, while LoFIT uses a fixed, manually defined subset.

**Data Efficiency.** Activation editing methods modify only a small fraction of the model's representational capacity, enabling strong performance in low-resource settings. *LoRA* performs competitively with as few as 1,000 training examples across various NLP benchmarks (Hu et al., 2021). For a detailed comparison of data efficiency between LoRA and JOLA, see Figure 8.

**Robustness.** Performance degradation due to sensitivity to hyperparameters or intervention configuration is a common concern. *LoRA* is relatively robust, as its low-rank formulation transfers well across domains, though it still requires manual tuning of the rank parameter. *Traditional activation editing* (e.g., LoFIT) depends on manual head selection and tuning, and shows high variance—up to ±5% accuracy—across datasets. **JOLA** addresses this by eliminating manual gate thresholding. Instead, it optimizes HardConcrete parameters end-to-end, allowing the model to dynamically select relevant heads per input or task. This results in consistently stable performance across diverse benchmarks (Table 1). We report baseline hyperparameter sensitivity results in Appendix F.3.

**Intervention Granularity.** The location and granularity of interventions affect both expressiveness and computational cost. We empirically compare different intervention components in Section 3.1 (**Q1**) and observe that attention heads are particularly crucial for task adaptation relative to other components. This insight directly motivates the architectural design of JOLA.

*Table 3.* **Comparison of full parameter tuning, LoRA, and activation editing methods.** We compare the intervention components, the percentage of new parameters introduced, the data efficiency, and robustness across different tasks. We use LLaMA-3 (Dubey et al., 2024) to compute the parameters. Note that full parameter tuning* does not introduce new parameters.

| | Trainable Params (%) | Active Params (%) | Intervention | Dynamic Localization? | Data Efficient? | Robust? |
|---|---|---|---|---|---|---|
| Full Parameter Tuning* | 100% | 100% | - | - | No | No |
| LoRA (Hu et al., 2021) | 0.2605% | 0.2605% | - | - | No | No |
| BitFit (Ben Zaken et al., 2022) | 0.0800% | 0.0800% | Bias Term | No | No | No |
| RED (Wu et al., 2024a) | 0.0040% | 0.0040% | MLP Layer | No | Yes | No |
| ReFT (Wu et al., 2024b) | 0.0300% | 0.0300% | Hidden Representation | No | Yes | No |
| LoFIT (Yin et al., 2024) | 0.0035% | 0.0002% | Attention | No | Yes | No |
| JOLA | 0.0065% | 0.0002% | Attention | Yes | Yes | Yes |

*Table 4.* Comprehensive comparison between LOFIT and JOLA.

| Aspect | LOFIT | JOLA |
|---|---|---|
| **Localization** | Two-stage process: (1) Select heads via learning multiplicative interventions; (2) Discard scaling, freeze heads, and train additive bias vectors. | Joint optimization: dynamically selects heads while learning interventions. |
| **Intervention Type** | Multiplicative and additive, but seperately. | Hybrid: additive biases + multiplicative scaling via adaptive gating. |
| **Sparsity Control** | L1 regularization on scaling factors; top-K head selection. | Hard Concrete gates with expected-L0 regularization; differentiable pruning. |
| **Flexibility** | Fixed intervention type (bias) post-localization. | Learns task-specific intervention type per head. |

# B. Comparative Analysis between JOLA and LoFIT

To further contextualize the contributions of JOLA, we provide a detailed comparison with LoFIT (Yin et al., 2024), focusing on three key dimensions: methodology, intervention formulation, and empirical performance.

**Methodological Comparison.** A central distinction between JOLA and LoFIT lies in their treatment of *localization* and *intervention optimization*. While LoFIT employs a two-stage pipeline, JOLA integrates both processes into a unified end-to-end framework. This design allows JOLA to dynamically adapt head selections and intervention types based on downstream task requirements, thereby avoiding the suboptimal decoupling present in LoFIT. We present the difference between these two methods in Table 4.

**Formula-Level Comparison.** At the operational level, JOLA generalizes LoFIT's additive-only formulation by enabling hybrid interventions per head. The following equations illustrate the key differences:

- LOFIT (Additive bias):

$$z_t^{(l,i)} \leftarrow z_t^{(l,i)} + v^{(l,i)} \tag{7}$$

    – This static form is limited to linear shifts of activations and may not suffice for nuanced task demands requiring amplification or suppression.

- JOLA (Hybrid intervention):

$$z_t^{(l,i)} \leftarrow \underbrace{(1 + g_m^{(l,i)} \cdot m^{(l,i)})}_{\text{Scaling}} \odot z_t^{(l,i)} + \underbrace{g_a^{(l,i)} \cdot a^{(l,i)}}_{\text{Bias}} \tag{8}$$

    – This hybrid approach allows both multiplicative and additive adjustments to token-level activations, providing more expressive control over model behavior. Task-specific gating ($g_m^{(l,i)}, g_a^{(l,i)}$) enables adaptive modulation per head.

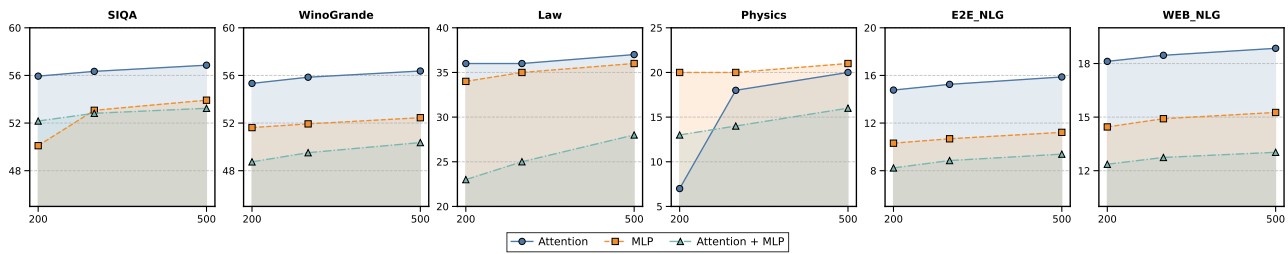

*Figure 10.* Performance comparison of interventions across different Transformer components and training sample sizes.

**Empirical Observations.**    JOLA demonstrates robust performance across 26 NLP tasks, particularly under low-resource settings. In addition to improved accuracy, JOLA is more parameter-efficient than LoFIT, due to shared head/intervention parameters and a learned gating mechanism that selects to not edit the unnecessary heads.

## C. Additional Analysis on Component Selection

To further investigate the impact of intervening on different Transformer components, we evaluate how performance scales across various data sizes (in low-resource settings) and component combinations.

**Degradation from Combining Interventions.**    As mentioned in Section 3.1 (**Q1**), combining interventions—particularly between attention heads and MLPs—tends to degrade performance. This is not simply a consequence of limited data. As shown in Figure 10, even with larger training sets (up to 500 examples), the combined Attention+MLP interventions underperform compared to using attention alone with fewer examples (e.g., 200). This pattern suggests a form of overfitting or representational interference when editing multiple components simultaneously. While MLP layers contribute to intermediate representation refinement, their effects do not appear to be complementary when naively combined with attention-level interventions.

**Performance Scaling with Sample Size.**    We also examine how performance changes as the amount of training data increases. While attention-only interventions benefit slightly from more data (as expected), the gap between attention and MLP-only interventions remains consistently large across all sample sizes. More notably, increasing data for the combined MLP+Attention configuration fails to close this gap, further emphasizing the importance of careful component selection rather than relying on brute-force data scaling.

These findings reinforce our conclusion: while multiple Transformer components are modifiable, not all interventions contribute equally. Moreover, indiscriminately editing more components can be detrimental. Attention heads remain the most effective target for activation-based editing, and intervention strategies should prioritize precision over breadth.

## D. Datasets

We conduct experiments across three tasks: commonsense reasoning (Hu et al., 2023), natural language understanding (Wang et al., 2024b), and natural language generation (Gehrmann et al., 2022). Table 5 provides a brief overview of the sub-datasets or sub-tasks within the three benchmarks evaluated. The commonsense reasoning task is framed as a multiple-choice problem, where the correct answer is selected from 2 to 4 possible options. The natural language understanding task also follows a multiple-choice format, but with ten options. The natural language generation task, on the other hand, is an end-to-end text generation task, where unstructured data (such as commonsense concepts or data) is converted into coherent text. In the training phase, we simulate a low-resource scenario by using 200 examples. Section 6.5 further explores experiments with varying numbers of samples. To ensure consistency across experiments, we used the same random seed (seed= 42) for data sampling, ensuring identical training samples in all runs.

## E. Prompt Setting

Recent studies (He et al., 2024; Lai et al., 2024) have highlighted the substantial impact of prompt design on model performance. In our experiments, we adopt the same prompt configurations as Hu et al. (2023) for the commonsense

*Table 5.* Overview of the sub-datasets and sub-tasks evaluated across three main tasks: commonsense reasoning, natural language under-standing, and natural language generation. Each task is designed to assess different aspects of language processing, with commonsense reasoning and natural language understanding framed as multiple-choice problems, and natural language generation as an end-to-end text generation task.

| Task | Dataset | Description | Label |
|---|---|---|---|
| Commonsense Reasoning (Hu et al., 2023) | ARC-c | Designed to challenge co-occurrence methods, similar to ARC-e but more complex. | answer1/answer2/answer3/answer4 |
| | ARC-e | Authentic grade-school level multiple-choice science questions. | answer1/answer2/answer3/answer4 |
| | BoolQ | A dataset for answering naturally occurring yes or no questions. | true/false |
| | HellaSwag | Select the most appropriate ending or sentence completion given a context. | ending1/ending2/ending3/ending4 |
| | OBQA | An open-book QA dataset requiring extensive knowledge. | answer1/answer2/answer3/answer4 |
| | PIQA | Focuses on physical commonsense reasoning. | solution1/solution2 |
| | SIQA | Involves reasoning about human actions and their social consequences. | answer1/answer2/answer3 |
| | WinoGrande | Fill-in-the-blank task with binary options within a sentence. | option1/option2 |
| MMLU-Pro (Wang et al., 2024b) | Biology Business Chemistry Computer Science Economics Engineering Health History Law Math Other Philosophy Physics Psychology | A question-answering task spanning 14 domains, primarily from the MMLU benchmark (Hendrycks et al., 2020), with additional examples from STEM resources[6]. | option1/option2/option3/option4/ option5/option6/option7/option8/ option9/option10 |
| GEM (Gehrmann et al., 2022) | Common_Gen | Converts concepts into coherent sentences. | No label, end-to-end text generation |
| | E2E_Nlg | Transforms structured data into natural language text. | |
| | Web_Nlg | Generates text from structured data inputs. | |
| | Xsum | Performs abstractive summarization of documents. | |

*Table 6.* Prompt settings are employed across various benchmarks, including Commonsense Reasoning, MMLU-Pro, and GEM.

| Benchmark | Task | Prompt |
|---|---|---|
| Commonsense Reasoning (Hu et al., 2023) | all 8 tasks | Please choose the correct answer to the question:{Question}. \n\n Option1:{option1}...Option4:{option4}\n\n Answer format: Option1/...\Option4. |
| MMLU-Pro (Wang et al., 2024b) | all 14 domains | The following are multiple choice questions (with answers) about {domain}. Please return the answer in the format of [The answer is (X)] at the end. Question: {Question} Options: A. {optionA}. B. {optionB}... J. {optionJ}. |
| GEM (Gehrmann et al., 2022) | Common_Gen | Ignoring the order of the concepts: {concepts}; \nGenerate a sentence with all the concepts. |
| | E2E_NLG | Please generate a restaurant description from the information given below:\n\nn{data} |
| | WEB_NLG | Take the following triple set as part of a Data-to-Text task: {data}. Make a lexicalization of the triple set into plain text. |
| | Xsum | First, please read the article below.\n\n{article}\n\nNow, can you write me an extremely short abstract for it? |

*Table 7.* Hyperparameter configurations for the baseline methods evaluated in our experiments. These settings are used across multiple tasks to assess model performance in low-resource settings, as discussed in Section 1 and Section 4.

| Baseline | Hyperparameter | Values |
|---|---|---|
| BitFit (Ben Zaken et al., 2022) | Bias Moudule | bias of Q,K and V from attention/bias of LayerNorm from attention outputs/bias of LayerNorm from hidden outputs |
| | Learning Rate | 1e-4/ 5e-4 |
| RED (Wu et al., 2024a) | Rank | 8 / 16 |
| | Learning Rate | 5e-5/ 2e-4 / 6e-2 |
| REPE (Zou et al., 2023) | method | Representation Reading / Representation Control |
| ReFT (Wu et al., 2024b) | Prefix + suffix posotion | p7 + s7 / p11 + s11 |
| | Layers | all / 4,6,10,12,14,18,20,22/3,9,18,24 |
| LoFIT (Yin et al., 2024) | number of attention heads | 32/64/128 |
| | Learning Rate | 5e-4 / 5e-3 |

reasoning benchmark, and used the prompts from the original paper for the MMLU-Pro benchmark (Wang et al., 2024b). For the GEM benchmark (Gehrmann et al., 2022), where the original paper did not provide the prompt settings, we utilized commonly used prompts curated from PromptSource[7]. To ensure reproducibility of our results, we present the prompts employed in our experiments in Table 6.

*Table 8.* Performance comparison of different learning rate (LR) schedules across six tasks for both JOLA and LoFIT models.

| | JOLA | | | | | | LoFIT | | | | | |
|---|---|---|---|---|---|---|---|---|---|---|---|---|
| | SIQA | WinoGrande | Law | Physics | E2E_NLG | WEB_NLG | SIQA | WinoGrande | Law | Physics | E2E_NLG | WEB_NLG |
| Linear | 62.71 | 56.49 | 38.00 | 42.00 | 14.05 | 22.83 | 54.13 | 53.36 | 35.00 | 6.00 | 13.84 | 16.95 |
| Cycle | 64.25 | 57.26 | 39.00 | 43.00 | 14.37 | 23.44 | 54.32 | 54.25 | 34.00 | 6.00 | 14.37 | 17.83 |
| Adaptive | 65.47 | **58.60** | 39.00 | 44.00 | 15.02 | 23.86 | 55.18 | **55.57** | **36.00** | 7.00 | **15.24** | 17.64 |
| Exponential | 66.22 | 58.33 | **40.00** | **46.00** | **15.54** | **24.39** | **55.94** | 55.33 | **36.00** | 7.00 | 14.77 | **18.12** |

# F. Experiment Configurations

### F.1. Training Setup

We conduct all experiments using the HuggingFace Transformers[8] library and fine-tuned the models with the TRL toolkit[9]. The AdamW optimizer (Loshchilov, 2017) was used for fine-tuning, with $\epsilon = 1e-6$ and one epoch of warm-up. Given the small dataset (e.g., 200 samples in our setting), overfitting was a concern. To mitigate overfitting's impact on the baseline, we introduced early stopping, which was not applied in the original implementation of the baseline systems. We also found that learning rate adjustment significantly affected the results. To optimize the learning rate strategy, we evaluated four strategies: (1) linear schedule (Mnih et al., 2015), (2) Cyclic Learning Rate Schedule (Smith, 2017), (3) Adaptive Heuristic Schedule (Smith, 2018), and (4) Exponential Decay Schedule (Li & Arora, 2019). As shown in Table 8, the exponential decay strategy proved to be the most stable, so we used it for both the baseline and our method. The exponentially decaying learning rate schedule is defined by the following formula:

$$\text{lr}(t) = \text{lr}_0 \cdot \lambda^t \cdot e^{-\text{decay} \cdot t} \tag{9}$$

where $\text{lr}_0$ is the initial learning rate $lr_0$ set to $5 \times 10^{-4}$, $\lambda$ is 0.1, and the decay rate is 0.01.

For the gating units, we used a temperature of 0.33 in the Gumbel Softmax (Jang et al., 2017). Fine-tuning was performed in full precision for the 7B, 8B, 1B, and 3B models, while for the 70B model, we applied 4-bit quantization to enable

---

[7]https://github.com/bigscience-workshop/promptsource
[8]https://github.com/huggingface/transformers
[9]https://github.com/huggingface/trl

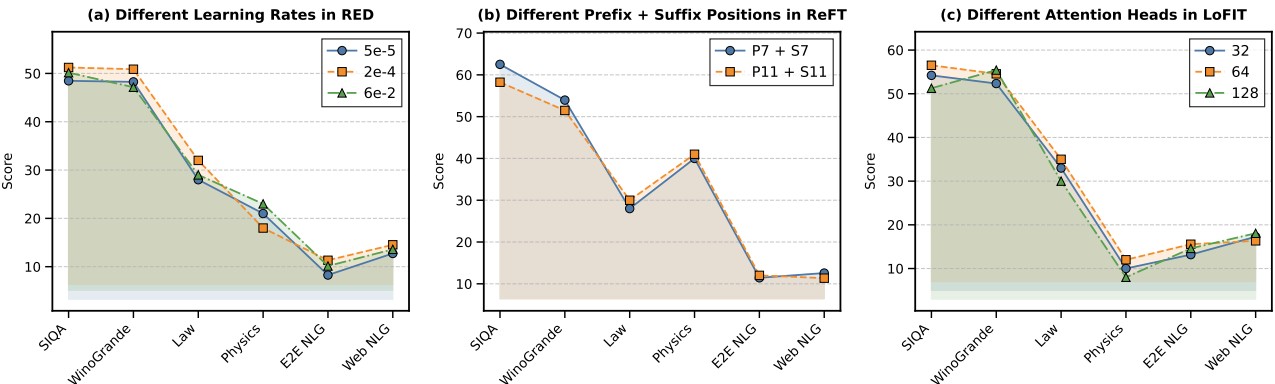

*Figure 11.* Performance comparison across six tasks under different experimental settings. The three subplots illustrate the sensitive of various configurations on task performance: (a) Different learning rates in RED (5e-5, 2e-4, 6e-2), (b) Different prefix and suffix positions in ReFT (P7 + S7, P11 + S11), and (c) Different numbers of attention heads in LoFIT (32, 64, 128).

single-precision fine-tuning.

### F.2. Computational Resources

All experiments for the 1B, 3B, 8B, and 13B models were conducted on a single NVIDIA A100 80GB GPU server. The 70B model, described in Section 6.5, was evaluated on an NVIDIA H100 94GB GPU server. As an example, with the 8B LLaMA-3 model, JOLA converged within 2 GPU hours on most tasks in the low-resource setting, using only 200 training samples.

### F.3. Hyperparameter Search for Baselines

As discussed in Section 1 and Section 4, the performance of baseline methods in low-resource settings is highly sensitive to hyperparameters across different tasks. We present in Figure 11 the sensitivity of hyperparameters in baseline methods, including the effects of varying learning rates in RED (Wu et al., 2024a), different prefix and suffix positions in ReFT (Wu et al., 2024b), and the number of attention heads in LoFIT (Yin et al., 2024). However, it is impractical to conduct hyperparameter searches for each task individually, given that we evaluate 26 tasks in total, and performing a separate search for each would be time-consuming. To mitigate this, we perform hyperparameter selection using a grid search approach. For each task, we run a grid search with five different hyperparameter configurations, which are chosen to explore a diverse range of parameter settings that could provide the best model performance. We performed this search over key hyperparameters, as presented in Table 7, using a validation set to select the configuration that resulted in the best performance. The final model is evaluated with these hyperparameters, and we averaged the results across all tasks, as reported in Table 1 and Figure 4.

## G. Full Results Across all Tasks

Due to page limitations, we present the average performance across the 26 tasks in Table 1 and Figure 4. In this section, we provide detailed performance metrics for each individual task. Specifically, Table 9 reports the accuracy of LLaMA-3 on the commonsense reasoning task, while Table 10 presents the accuracy of Qwen-2.5 on the same task. Table 11 shows the accuracy of LLaMA-3 on the natural language understanding task, and Table 12 shows the corresponding accuracy for Qwen-2.5. Finally, Table 13 presents the BLEU, ROUGE-L, and BERTScore for LLaMA-3 on the natural language generation task, with Table 14 displaying the corresponding metrics for Qwen-2.5.

## H. Case Study

To provide an intuitive evaluation of the advantages of our method, we select one representative case from each of the tasks: commonsense reasoning, natural language understanding, and natural language generation. The results generated by the baseline and our approach are presented below.

*Table 9.* The accuracy of LLaMA-3 across various commonsense reasoning tasks, comparing different baseline methods and our proposed method (JOLA).

| | ARC-c | ARC-e | BoolQ | HellaSwag | OBQA | PIQA | SIQA | WinoGrande | # AVG |
|---|---|---|---|---|---|---|---|---|---|
| **zero_shot** | 59.56 | 65.40 | 41.99 | 45.19 | 54.80 | 76.01 | 42.78 | 43.88 | 53.70 |
| **LoRA** (Hu et al., 2021) | 70.13 | 77.85 | 56.37 | 66.18 | 73.38 | 71.36 | 63.42 | 53.97 | 66.58 |
| **BitFit** (Ben Zaken et al., 2022) | 64.17 | 72.35 | 49.69 | 63.48 | 74.07 | 71.24 | 58.14 | 51.28 | 63.05 |
| **RED** (Wu et al., 2024a) | 39.67 | 56.20 | 9.69 | 40.81 | 50.75 | 70.09 | 50.46 | 51.84 | 46.19 |
| **RePE** (Zou et al., 2023) | 61.34 | 74.07 | 53.41 | 60.32 | 76.06 | 73.18 | 60.53 | 49.95 | 63.61 |
| **ReFT** (Wu et al., 2024b) | 66.36 | 77.37 | 53.34 | 63.96 | 75.43 | 73.50 | 62.27 | 55.36 | 65.95 |
| **LoFIT** (Yin et al., 2024) | 57.10 | 78.59 | 43.69 | 42.01 | 61.73 | 53.96 | 56.37 | 56.10 | 56.19 |
| **JOLA** | **74.66** | **80.13** | **62.17** | **70.69** | **76.20** | **76.01** | **66.22** | **58.33** | **70.55** |

*Table 10.* The accuracy of Qwen-2.5 across various commonsense reasoning tasks, comparing different baseline methods and our proposed method (JOLA).

| | ARC-c | ARC-e | BoolQ | HellaSwag | OBQA | PIQA | SIQA | WinoGrande | # AVG |
|---|---|---|---|---|---|---|---|---|---|
| **zero_shot** | 88.14 | 94.70 | 55.87 | 82.42 | 81.80 | **87.38** | **76.56** | 62.35 | 78.65 |
| **LoRA** (Hu et al., 2021) | 85.30 | 92.04 | 66.20 | 83.30 | 82.09 | 84.53 | 71.25 | 61.53 | 78.28 |
| **BitFit** (Ben Zaken et al., 2022) | 75.73 | 85.78 | 53.05 | 75.37 | 70.08 | 78.63 | 66.08 | 49.23 | 69.25 |
| **RED** (Wu et al., 2024a) | 84.23 | 86.72 | 55.74 | 75.09 | 73.05 | 79.00 | 67.28 | 51.08 | 71.52 |
| **RePE** (Zou et al., 2023) | 78.90 | 83.51 | 55.49 | 74.18 | 68.38 | 81.45 | 63.20 | 53.72 | 69.85 |
| **ReFT** (Wu et al., 2024b) | 79.29 | 87.57 | 58.88 | 77.72 | 71.96 | 82.41 | 69.66 | 54.01 | 72.69 |
| **LoFIT** (Yin et al., 2024) | 78.32 | 84.25 | 54.14 | 75.00 | 72.53 | 79.27 | 66.60 | 49.30 | 69.93 |
| **JOLA** | **88.31** | **95.29** | **68.10** | **88.53** | **86.40** | 87.05 | 75.79 | **69.69** | **82.40** |

*Table 11.* The performance of LLaMA-3 across multiple domains in the MMLU-Pro benchmark.

| | Biology | Business | Chemistry | Computer Science | Economics | Engineering | Health | History | Law | Math | Other | Philosophy | Physics | Psychology | #AVG |
|---|---|---|---|---|---|---|---|---|---|---|---|---|---|---|---|
| **zero_shot** | **75** | 31 | 27 | **36** | 53 | 26 | 55 | 46 | 31 | 20 | 34 | 43 | 24 | 59 | 40 |
| **LoRA** | 65 | 35 | 35 | 30 | 47 | 40 | 54 | 51 | 36 | 31 | 32 | 36 | 44 | 53 | 42 |
| **BitFit** | 62 | 31 | 20 | 30 | 49 | 23 | 47 | 43 | 20 | 22 | 24 | 45 | 21 | 53 | 35 |
| **RED** | 59 | 26 | 26 | 31 | 52 | 26 | **61** | 46 | 34 | 23 | 33 | 42 | 21 | 41 | 37 |
| **RePE** | 67 | 30 | 22 | 30 | **56** | 26 | 49 | 37 | 24 | 14 | 24 | 35 | 28 | 55 | 36 |
| **ReFT** | 71 | 31 | 31 | 35 | 57 | 26 | 55 | 45 | 31 | 23 | 32 | **46** | 29 | 61 | 41 |
| **LoFIT** | 55 | 17 | 13 | 29 | 37 | 32 | 29 | 49 | 37 | 16 | 19 | 16 | 7 | 33 | 28 |
| **JOLA** | 70 | **42** | **43** | 34 | 53 | **43** | 55 | **54** | 40 | 37 | 40 | 39 | **46** | 62 | **47** |

*Table 12.* The performance of Qwen-2.5 across multiple domains in the MMLU-Pro benchmark.

| | Biology | Business | Chemistry | Computer Science | Economics | Engineering | Health | History | Law | Math | Other | Philosophy | Physics | Psychology | #AVG |
|---|---|---|---|---|---|---|---|---|---|---|---|---|---|---|---|
| **zero_shot** | 71 | 20 | 17 | 36 | 55 | 17 | 46 | 44 | 27 | 13 | 44 | 48 | 19 | 64 | 37 |
| **LoRA** | 68 | 32 | 36 | 45 | 58 | 36 | 48 | 53 | 34 | 40 | 33 | 40 | 50 | 74 | 46 |
| **BitFit** | 49 | 25 | 13 | 17 | 40 | 26 | 25 | 30 | 9 | 18 | 25 | 29 | 29 | 66 | 29 |
| **RED** | 73 | 21 | 20 | 40 | 56 | 23 | 49 | 43 | 28 | 15 | **45** | 46 | 20 | 65 | 39 |
| **RePE** | 57 | 32 | 20 | 22 | 42 | 23 | 17 | 14 | 21 | 34 | 22 | 9 | 27 | 68 | 29 |
| **ReFT** | 74 | 38 | 37 | 46 | 55 | 29 | 53 | 50 | 37 | 36 | 42 | 44 | 54 | 74 | 48 |
| **LoFIT** | 73 | 25 | 23 | 43 | 58 | 35 | 51 | 47 | 29 | 27 | 44 | **49** | 31 | 67 | 43 |
| **JOLA** | **75** | **41** | **39** | **49** | **62** | **38** | **56** | **57** | **42** | **45** | 42 | 45 | **55** | **76** | **52** |

*Table 13.* The performance of LLaMA-3 across various natural language generation tasks (Commen_Gen, E2E_NLG, WEB_NLG, and Xsum), using BLEU, ROUGE-L, and BERTScore as evaluation metrics.

| | Commen_Gen | | | E2E_NLG | | | WEB_NLG | | | Xsum | | |
|---|---|---|---|---|---|---|---|---|---|---|---|---|
| | BLEU | Rouge-L | BertScore | BLEU | Rouge-L | BertScore | BLEU | Rouge-L | BertScore | BLEU | Rouge-L | BertScore |
| **zero_shot** | 16.19 | 46.59 | 79.69 | 8.26 | 27.47 | 74.10 | 21.65 | 52.11 | 83.79 | 4.14 | 20.65 | 71.35 |
| **LoRA** | 18.17 | 49.54 | 81.15 | 13.15 | 39.75 | 77.50 | 19.53 | 34.50 | 82.18 | 2.25 | 24.11 | 70.12 |
| **BitFit** | 13.16 | 31.02 | 77.51 | 9.25 | 31.28 | 74.77 | 12.25 | 40.25 | 76.86 | 2.35 | 12.68 | 70.19 |
| **RED** | 17.19 | 45.41 | 80.43 | 10.31 | 30.44 | 75.51 | 14.45 | 42.62 | 78.43 | 2.99 | 11.14 | 70.60 |
| **RePE** | 11.24 | 30.15 | 76.15 | 8.12 | 25.46 | 74.01 | 12.36 | 42.36 | 76.94 | 2.25 | 12.46 | 70.12 |
| **ReFT** | 20.22 | 48.26 | 82.69 | 12.60 | 32.71 | 77.11 | 13.09 | 43.36 | 77.46 | 4.49 | 23.22 | 71.58 |
| **LoFIT** | 12.17 | 30.53 | 76.81 | 14.77 | 38.88 | 78.66 | 18.12 | **46.38** | 81.11 | 2.47 | 12.57 | 70.26 |
| **Our** | **23.13** | **53.47** | **84.93** | **15.54** | **42.52** | **79.22** | **24.39** | 38.09 | **85.92** | **5.24** | **28.50** | **72.07** |

*Table 14.* The performance of Qwen-2.5 across various natural language generation tasks (Commen_Gen, E2E_NLG, WEB_NLG, and Xsum), using BLEU, ROUGE-L, and BERTScore as evaluation metrics.

| | Commen_Gen | | | E2E_NLG | | | WEB_NLG | | | Xsum | | |
|---|---|---|---|---|---|---|---|---|---|---|---|---|
| | BLEU | Rouge-L | BertScore | BLEU | Rouge-L | BertScore | BLEU | Rouge-L | BertScore | BLEU | Rouge-L | BertScore |
| **zero_shot** | 14.58 | 41.85 | 78.53 | 8.08 | 25.63 | 73.97 | 31.13 | 56.11 | 91.40 | 2.32 | 13.59 | 70.17 |
| **LoRA** | 17.16 | 52.39 | 80.40 | 23.39 | 46.65 | 85.14 | 31.00 | 55.50 | 91.30 | 6.29 | 26.84 | 72.77 |
| **BitFit** | 14.92 | 40.16 | 78.77 | 15.25 | 35.03 | 79.01 | 21.49 | 43.27 | 83.66 | 2.23 | 13.94 | 70.11 |
| **RED** | 13.91 | 41.75 | 78.04 | 8.25 | 26.55 | 74.09 | 26.58 | 54.64 | 87.67 | 2.50 | 16.06 | 70.28 |
| **RePE** | 11.46 | 41.01 | 76.31 | 13.11 | 30.46 | 77.47 | 21.94 | 46.25 | 84.01 | 2.26 | 14.58 | 70.13 |
| **ReFT** | 15.84 | 41.37 | 79.43 | 18.05 | 35.43 | 81.07 | 25.04 | 48.93 | 86.44 | 5.15 | 23.85 | 72.01 |
| **LoFIT** | 11.23 | 40.73 | 76.15 | 9.17 | 28.47 | 74.72 | 26.47 | 54.50 | 87.58 | 2.36 | 15.02 | 70.19 |
| **Our** | **21.12** | **57.54** | **83.38** | **28.32** | **52.60** | **89.08** | **35.32** | **58.54** | **94.99** | **11.24** | **32.25** | **76.15** |

---

**Case 1: Commonsense Reasoning Task (ARC-c) on LLaMA-3.1-8B**

**Instruction:** Please choose the correct answer to the question: {Question}\n\n Answer1: {answer1}\n Answer2: {answer2}\n Answer3: {answer3}\n Answer4: {answer4} \n\n Answer format: answer1/answer2/answer3/answer4\n\n ### Response:\n

**Question:** A definite shape and a definite volume are properties of which state of matter?

**Answers:** Answer1: solid, only\n Answer2: liquid, only\n Answer3: solid and liquid\n Answer4: liquid and gas\n

- - - - - - - - - - - - - - - - - - - - - - - - - - - - - - - - - - - - - - - -

**Zero-Shot**: solid, only/liquid, only/solid and liquid/liquid and gas. (Non Results)

**LoRA**: answer1 (True Answer)

**BitFit**: solid, only/liquid, only (Non Results)

**RED**: answer3 (Wrong Answer)

**REPE**: answer3 (Wrong Answer)

**ReFT**: answer3 (Wrong Answer)

**LoFIT**: answer4 (Wrong Answer)

**JoLA**: answer1 (True Answer)

---

Case 2: Natural Language Understanding Task (Math) on Qwen-2.5-7B

**Instruction:** The following are multiple choice questions (with answers) about {domain}. Please return the answer in the format of "The answer is (X)" at the end.\n Question: {question} \n Options: A. {A} B. {B} C. {C} D. {D} E. {E} F. {F} G. {G} H. {H} I. {I} J. {J}\n ### Response:\n
**Question:** Use Euler's Method to calculate the approximation of y(0.2) where y(x) is the solution of the initial-value problem that is as follows. $y'' + xy' + y = 0$ and $y(0) = 2$ and $y'(0) = 3$.
**Options:** A. 2.67 B. 2.33 C. 1.76 D. 1.98 E. 2.10 F. 2.58 G. 2.82 H. 3.14 I. 3.02 J. 2.45 \n

- - - - - - - - - - - - - - - - - - - - - - - - - - - - - - - - - - - - - - - -

**Zero-Shot**: E (Wrong Answer)
**LoRA**: E (Wrong Answer)
**BitFit**: B (Wrong Answer)
**RED**: E (Wrong Answer)
**REPE**: C (Wrong Answer)
**ReFT**: !!!!!"!!!!#!!!!$ (Non Results)
**LoFIT**: F (True Answer)
**JoLA**: F (True Answer)

---

Case 3: Natural Language Generation Task (E2E_NLG) on Qwen-2.5-7B

**Instruction:** Please generate a restaurant description from the information given below: {data} \n\n ### Response:\n
**Data:** name[Zizzi], eatType[pub], near[The Sorrento]
**Reference:** There is a pub called Zizzi located near The Sorrento.

- - - - - - - - - - - - - - - - - - - - - - - - - - - - - - - - - - - - - - - -

**Zero-Shot**: Nestled in the heart of the city, just a stone's throw away from The Sorrento, Zizzi is a charming and cozy pub that offers a delightful dining experience. With its warm and inviting atmosphere, Zizzi is the perfect spot for a casual meal, a relaxing drink, or a lively gathering ... (Tend to generate without end)
**LoRA**: Zizzi is a pub located near The Sorrento. It is a great place to grab a bite to eat and enjoy a drink in a lively atmosphere. Zizzi's menu features a variety of delicious dishes, from classic pub fare to more adventurous options. Whether you're in the mood for a hearty burger... (Tend to generate without end)
**BitFit**: There is a pub called Zizzi near The Sorrento. It is a good place to eat. It is a pub. It is near The Sorrento. It is called Zizzi. It is a good place to eat. It is a pub... (Tend to generate repetitions without end)
**RED**: Nestled in the heart of the city, just a stone's throw away from The Sorrento, Zizzi is a charming and cozy pub that offers a delightful dining experience. With its warm and inviting atmosphere, Zizzi is the perfect spot for a casual meal, a relaxing drink, or a... (Tend to generate without end)
**REPE**: Zizzi is a pub. It is located near The Sorrento. Zizzi is a place where you can go to eat. Zizzi is a place where you can go to drink. Zizzi is a place where you can go to socialize. Zizzi... (Tend to generate repetitions without end)
**ReFT**: Zizzi is a pub in The Sorrento. Zizzi is a restaurant in The Sorrentos. Zizzis is a restaurant in the sorrentos. The Zizzis is a restaurant at the sorrentos.. Zizzi's is a restaurant in sor... (Tend to generate repetitions without end)
**LoFIT**: Nestled in the heart of the city, just a stone's throw away from The Sorrento, Zizzi is a charming and cozy pub that offers a delightful dining experience. With its warm and inviting atmosphere, Zizzi is the perfect spot for a casual meal or a relaxing drink with friends. (Ends normally, but no Sorrento-related information.)
**JoLA**: Zizzi is a pub near The Sorrento. (True Answer)

