# OpenReview forum: "Joint Localization and Activation Editing for Low-Resource Fine-Tuning"
_ICML.cc/2025/Conference — ICML 2025 poster_

### Official Review · Reviewer_m3pE · 2025-03-13

**Overall Recommendation:** 2

**Summary:**

The paper proposes JOLA, a novel parameter-efficient fine-tuning (PEFT) method that dynamically selects and edits the outputs of specific Transformer attention heads. JOLA jointly learns: (1) which heads to target, (2) the intervention type (additive, multiplicative, or both), and (3) the corresponding parameters (additive offsets/multiplicative scalings). Evaluated on three NLP tasks—commonsense reasoning, language understanding, and generation—JOLA outperforms baselines under low-resource settings, demonstrating improved stability and efficiency through its unified framework of localization and modular activation editing.

**Claims And Evidence:**

Yes.

**Essential References Not Discussed:**

No.

**Experimental Designs Or Analyses:**

Yes, no issues.

**Methods And Evaluation Criteria:**

Yes.

**Other Comments Or Suggestions:**

1.	To enhance clarity in Figure 1, I suggest utilizing distinct colors for the boxes representing different methods. This would allow for easier differentiation and improve the overall visual appeal of the figure.
2.	In Table 2, it would be beneficial to bold the maximum value to draw attention to this important data point, thereby facilitating a quicker understanding of the results presented.

**Other Strengths And Weaknesses:**

Strengths:

1.	The article has a clear and coherent structure with well-organized content.
2.	The article provides rich visual and analytical insights. The figures and tables effectively clarify complex concepts. The visualization of gate status provides valuable empirical evidence of JOLA's mechanism.
3.	JOLA outperforms baselines across multiple NLP tasks, particularly under low-resource settings. The results are well-documented, and the performance gains are consistent, supporting the method's efficacy.

Weaknesses:

1.	In Figure 1, you can draw a red box structure diagram of LoFIT like RED and JOLA.
2.	In Figure 1, LoFIT's Step 2, it's a "+", not a "x" sign.
3.	The article mentions "However, the effectiveness of standard PEFT methods is limited in low-resource scenarios with only a few hundred examples." The PEFT method is a series of fine-tuning methods proposed to reduce computing resources. This turn of the article is to explain the shortcomings of PEFT, but the low-resource background is not a defect of PEFT itself. The logical relationship between this and the following text is not clear.
4.	I hope the article can provide a detailed comparative analysis between JOLA and LOFIT. From the current description of the article, LOFIT's m is learnable, but the maximum value of the value range is a hyperparameter and needs to be set manually. So, is LoFIT's m the product of JoLA's g and m? If LoFIT's m is 0, does it mean that the head is not selected? Similarly, is LoFIT's a the sum of JoLA's g and a? In addition, the article mentions that LoFIT determines the selected head by multiplication, but JOLA also uses multiplication to determine the selected head. What is the specific difference here? Or are the two stages of LOFIT not distributed as shown in Figure 1, because at present, the equation for synthesizing one stage is also acceptable.

**Questions For Authors:**

1.	It is worth exploring why JoLA incorporates two additional gates while maintaining the same overall number of parameters as LOFIT. A thorough explanation of this architectural choice would enhance the reader's understanding of how JoLA optimally manages its parameters.
2.	In Figure 2, the performance of editing attention modules on the Physics task appears to be the lowest. An analysis of this outcome is warranted.

**Relation To Broader Scientific Literature:**

While the paper introduces JOLA as a novel PEFT method for joint localization and activation editing, there is insufficient clarity in distinguishing it from LOFIT, a closely related prior work that also intervenes on specific Transformer heads with additive/multiplicative editing. Both JOLA and LOFIT aim to (1) identify select heads for editing and (2) apply combined additive/multiplicative interventions. The paper does not clearly articulate what differentiates JOLA’s joint optimization framework from LOFIT’s approach, although the performance of JOLA is higher than LOFIT.

**Theoretical Claims:**

No theoretical claims.

---

> ### Author Rebuttal · Authors · 2025-03-31
>
> Thank you for your insightful comments and suggestions
>
> **Q1: Presentation in Figure 1, Table 2.**
>
> **A1:** Thank you for your suggestion. We will make the necessary changes in the next version.
>
> **Q2: Relationship between PEFT?**
>
> **A2:** In the abstract, we introduced PEFT and its limitations in low-resource scenarios. Activation editing is a form of PEFT but updates fewer parameters compared to traditional methods like Adapter and LoRA. Traditional methods are less suitable for low-resource scenarios, and our approach aims to tackle this issue. We will clarify this logic in the next version to avoid any confusion.
>
> **Q3: Provide a detailed comparative analysis between JOLA and LOFIT.**
>
> **A3:** Below, we provide a  comparison focusing on methodological differences, intervention strategies, and empirical performance.
>
> (1) Methodological differences: Unlike LoFiT’s rigid two-stage pipeline, JoLA unifies localization and intervention into a single end-to-end framework. This allows dynamic adaptation to task requirements, avoiding suboptimal head selection caused by decoupled optimization.
> || LoFIT|JoLA|
> |-|-|-|
> |Localization|Two-stage process: 1. Select heads via learning with multiplicative interventions 2. Freeze selection and train additive interventions (bias vectors). | Joint optimization: Dynamically selects heads while learning intervention parameters.|
> |Intervention Type|Additive only (bias vectors applied to selected heads).|Hybrid intervention: Combines additive biases and multiplicative scaling via adaptive gating.|
> |Sparsity Control|L1 regularization on scaling factors to select top-K heads.|Hard Concrete gates with expected-L0 regularization, enabling probabilistic head pruning during training.|
> |Flexibility|Fixed intervention strategy (additive) after head selection.|Learns task-specific intervention types (e.g., scaling vs. bias) per head.|
>
> (2) Formula-level comparison
>
> **LoFIT:**
> $$z^{(l,i)}_t \leftarrow z^{(l,i)}_t + v^{(l,i)} \quad \text{(Additive bias)}$$
>
> **Limitation:** Static additive edits may insufficiently adapt complex tasks requiring both amplification and suppression of features.
>
> **JoLA:**
> $$
> z^{(l,i)}_t = (1 + g_m^{(l,i)} \cdot m^{(l,i)}) \odot z^{(l,i)}_t + g_a^{(l,i)} \cdot a^{(l,i)}
> $$
>
> where,
> Scaling: $(1 + g_m^{(l,i)} \cdot m^{(l,i)})$,
> Bias: $g_a^{(l,i)} \cdot a^{(l,i)}$
>
> **Advantage:** The hybrid operation enables fine-grained control over activations—scaling amplifies/suppresses existing features, while biases shift representations.
>
> (3) Empirical Performance
> + JoLA show robustness across 26 tasks under low-resource settings.
> + JoLA is more parameter efficient than LoFIT, we will update the parameter counting as discussed in Review 2Bjt [Q2] in the new version of the paper.
>
> **Q4: Parameter issues**
>
> **A4:** As noted in the original papers for each baseline method, these methods are sensitive to hyperparameters, and we detail this in Appendix D.3. The following table shows the sensitivity of these methods, with the results highlighting the lack of robustness, which motivates our proposal for a more stable approach.
>
> **Different learning rate in RED**
> ||SIQA|WinoGrande|Law|Physics|e2e_nlg|Web_nlg|
> |-|-|-|-|-|-|-|
> |5e-5|48.47|48.25|28.00|21.00|8.27|12.74|
> |2e-4|51.23|50.87|32.00|18.00|11.33|14.53|
> |6e-2|50.16|47.14|29.00|23.00|10.14|13.59|
> **Different prefix and suffix positions in ReFT**
> ||SIQA|WinoGrande|Law|Physics|e2e_nlg|Web_nlg|
> |p7+s7|62.51|53.96|28.00|40.00|11.44|12.57|
> |p11+s11|58.24|51.48|30.00|41.00|12.02|11.35|
> **Number of Attention Heads in LoFIT**
> ||SIQA|WinoGrande|Law|Physics|e2e_nlg|Web_nlg|
> |32|54.21|52.34|33.00|10.00|13.18|17.27|
> |64|56.52|54.56|35.00|12.00|15.56|16.34|
> |128|51.26|55.42|30.00|8.00|14.60|18.07|
>
> **Q5: What happens on the physics task in Figure 2?**
>
> **A5:** Upon reviewing the generated text for the physics task in Figure 2, we found that in the low-resource setting, the generated outputs did not consistently follow the required format, which impacted the results. Specifically, the generated text lacked the necessary result options, leading to lower accuracy when evaluated using exact match criteria.

---

### Official Review · Reviewer_upuz · 2025-03-13

**Overall Recommendation:** 3

**Summary:**

This paper introduces JOLA, a novel approach to efficiently adapt large language models in low-resource settings. The method jointly learns which attention heads to modify and determines optimal activation interventions using both additive and multiplicative adjustments. By incorporating a dynamic gating mechanism based on Hard-Concrete distributions with expected-L0 regularization, JOLA selectively edits a minimal set of components, thereby significantly reducing the number of trainable parameters. Extensive experiments on tasks spanning commonsense reasoning, natural language understanding, and language generation demonstrate that JOLA consistently outperforms existing methods such as LoRA and other activation editing techniques, achieving robust and scalable improvements even with limited training data.

**Claims And Evidence:**

The claims proposed in the paper are substantiated by extensive experimental evidence, demonstrating that the JOLA method not only consistently outperforms baseline activation editing and parameter-efficient fine-tuning techniques in low-resource scenarios, but also significantly enhances performance across a diverse set of tasks including commonsense reasoning, natural language understanding, and generation. Moreover, the ablation studies provide clear and convincing evidence that the dynamic gating mechanism and selective attention head modifications are key factors driving these improvements, effectively validating the paper's central claims.

**Essential References Not Discussed:**

na

**Experimental Designs Or Analyses:**

The paper presents a thorough experimental design to evaluate the proposed Joint Localization and Activation Editing (JOLA) method for low-resource fine-tuning. In addition, the inclusion of extensive ablation studies (on gate mechanisms, the number of gates, and head selection strategies) and analyses across different data and model sizes strengthens the validity of their claims. The author mentioned that there is a huge difference in hyperparameter selection among the baseline methods. Could the extreme value variance for JORA and the other methods be displayed? Furthermore, providing a more detailed explanation of the statistical significance of the results and the reproducibility of the experiments would strengthen confidence in the research findings.

**Methods And Evaluation Criteria:**

The Joint Localization and Activation Editing (JOLA) approach presented in this paper is soundly designed. The challenge of efficiently adapting large language models with scarce data stems from the difficulty in discerning which internal components yield the greatest impact. JOLA overcomes this by simultaneously determining the most influential attention mechanisms to modify and selecting the appropriate adjustment strategy—be it scaling, shifting, or a combination—while fine-tuning the corresponding parameters, thereby ensuring that the targeted modifications are optimally calibrated for low-resource environments.

**Other Comments Or Suggestions:**

na

**Other Strengths And Weaknesses:**

Strengths:
Originality:
JOLA introduces an innovative approach by jointly determining which attention heads to modify and optimizing both scaling and shifting interventions, thereby creatively combining insights from activation editing and network pruning.
Significance:
The extensive experiments across commonsense reasoning, language understanding, and generation tasks clearly demonstrate that JOLA substantially outperforms existing fine-tuning methods in low-resource settings, underscoring its potential impact on efficient model adaptation.
Clarity:
The paper is well-structured and offers detailed explanations of the dynamic gating mechanism and head selection strategy, with comprehensive ablation studies that effectively illustrate the contribution of each component.

Weaknesses:
The approach’s reliance on specific gating mechanisms and parameter settings may introduce additional complexity and sensitivity to hyperparameter tuning, potentially affecting reproducibility.

**Questions For Authors:**

1.	How does the JOLA method perform when given larger resources? Can this activation-based editing approach be applied to larger datasets?
2.	The paper shows that the activation editing method performs best when applied to the Attention layer. Is this also the case for JOLA? Could the effectiveness of JOLA on the MLP layer be verified?

**Relation To Broader Scientific Literature:**

To improve low-resource adaptation of large language models, prior works have explored both parameter-efficient fine-tuning methods, such as LoRA and BitFit, and activation editing techniques like RED, REPE, and LoFIT. The JoLA method proposed in this paper innovatively combines dynamic attention head localization with joint optimization of additive and multiplicative interventions. Drawing on ideas from network pruning—specifically, HardConcrete gating and expected-L0 regularization—JoLA overcomes the static limitations of previous methods, resulting in a more adaptive and robust fine-tuning strategy that aligns well with contemporary advances in model adaptation.

**Theoretical Claims:**

The paper does not include theoretical proofs.

---

> ### Author Rebuttal · Authors · 2025-03-31
>
> We appreciate your comments and the opportunity to clarify and improve our work.
>
> Q1: Huge difference in hyperparameter selection among the baseline methods and JoLA. The reproducibility of our proposed JoLA
>
> A1: As noted in the original papers for each baseline method, these methods are sensitive to hyperparameters, and we detail this in Appendix D.3. The following table shows the sensitivity of these methods, with the results highlighting the lack of robustness, which motivates our proposal for a more stable approach.
>
> **Different learning rate in RED**
> ||SIQA|WinoGrande|Law|Physics|e2e_nlg|Web_nlg|
> |-|-|-|-|-|-|-|
> |5e-5|48.47|48.25|28.00|21.00|8.27|12.74|
> |2e-4|51.23|50.87|32.00|18.00|11.33|14.53|
> |6e-2|50.16|47.14|29.00|23.00|10.14|13.59|
> **Different prefix and suffix positions in ReFT**
> ||SIQA|WinoGrande|Law|Physics|e2e_nlg|Web_nlg|
> |p7+s7|62.51|53.96|28.00|40.00|11.44|12.57|
> |p11+s11|58.24|51.48|30.00|41.00|12.02|11.35|
> **Number of Attention Heads in LoFIT**
> ||SIQA|WinoGrande|Law|Physics|e2e_nlg|Web_nlg|
> |32|54.21|52.34|33.00|10.00|13.18|17.27|
> |64|56.52|54.56|35.00|12.00|15.56|16.34|
> |128|51.26|55.42|30.00|8.00|14.60|18.07|
>
> Q2: Providing a more detailed explanation of the statistical significance of the results.
>
> A2: Thank you for your suggestion. We will incorporate the statistical significance; note though that the gap is large
> We agree that conducting a statistical significance analysis is important to strengthen the credibility of our results. The significance test shows a meaningful difference between our method and the baseline methods.
>
> Q3: How does the JoLA method perform when given larger resources?
>
> A3: Thank you for your suggestion. We conducted experiments on larger datasets, and as shown in the table, our method remains effective with 5,000 and 10,000 samples. However, using 20,000 and 100,000 samples, there is a slight gap between our method and LoRA. These differences are acceptable, as we update fewer parameters compared to LoRA.
>
> ||SIQA||WinoGrande||
> |-|-|-|-|-|
> ||JoLA|LoRA|JoLA|LoRA|
> |1000|74.77|70.04|74.30|71.16|
> |2000|75.24|73.12|74.92|71.75|
> |3000|75.43|73.85|75.37|72.32|
> |5000|75.88|74.35|75.91|73.57|
> |8000|75.91|75.14|76.20|74.69|
> |10000|75.96|75.69|76.31|75.92|
> |20000|76.02|76.48|76.55|76.47|
> |30000|76.08|76.94|76.68|77.24|
> |50000|76.15|77.23|76.84|78.33|
> |80000|76.21|77.56|76.92|78.52|
> |100000|76.26|77.81|77.04|78.96|
>
> Q4: Could the effectiveness of JoLA on the MLP layer be verified?
>
> A4: We also applied the gating mechanism-based strategy to the MLP layer. As shown in the following table, our experimental results demonstrate that the JoLA method is also effective in the MLP layer. However, it empirically tends to be more effective when used in the attention layer.
>
> ||Reasoning||Understanding||Generation||
> |-|-|-|-|-|-|-|
> ||SIQA|WinoGrande|Law|Physics|e2e_nlg|Web_nlg|
> |MLP w/o gate|50.10|51.62|34.00|20.00|10.31|14.45|
> |MLP with gate|52.46|52.43|36.00|23.00|11.23|16.25|
> |Attention w/o gate|55.94|55.33|36.00|7.00|14.77|18.12|
> |Attention with gate|66.22|58.33|40.00|46.00|15.54|24.39|
> |Attention+MLP w/o gate|52.17|48.74|23.00|13.00|8.23|12.36|
> |Attention+MLP with gate|53.28|52.07|27.00|16.00|10.42|14.83|

---

### Official Review · Reviewer_2Bjt · 2025-03-13

**Overall Recommendation:** 3

**Summary:**

The paper proposes JORA, an interpretability-inspied parameter-effecient tuning methods. JOLA intervenes on the attention activations with both scaling and offsetting. In addition, JOLA uses HardConcrete gates with expected-L0 regularization to learns the localization together with intervention in an end to end way.

The paper conducts evaluation on multiple datasets and mainly compares against other activation editing baselines (ReFT and LoFiT). The results suggest the proposed approach is effective.

## update after rebuttal

Thank you for your response. I've update the score.

**Claims And Evidence:**

The claims are mostly valid. JoLA extends previous methods by jointly learning to localize and edit activations, which seems effective. The paper also provides abundant analysis.

Though I have some concerns in the way that the paper sets up the comparison with respect to the baselines, and the way that the paper discusses the number or parameters.

**Essential References Not Discussed:**

Not that I am aware of.

**Experimental Designs Or Analyses:**

First, I have some concerns about the way that the paper sets up the baselines.

The Baseline results seem either under-tuned or too weak to compare with: In Table 1, a lot of the fine-tuning baselines underperform zero-shot (e.g. for Qwen2.5 7B on the reasoning tasks, none of the 6 baselines outperform the zero-shot baseline).

I wonder how the bae line is set up. The paper claims that baseline results were obtained by "selecting five hyperparameters and **averaging** the results." (ln 246) A more common approach is establishing the best results out of all the five hyperparameters as the baselines.

Second, I wonder how is the number of learnable parameters counted in Table 3. Is it only counting the gating variables, or is it counting the "activated" attention heads? If I understand correctly, JOLA incorporates two gating, one scaling, and one additive parameters for each attention heads. All these are learned end to end and need **gradient back-propagation**. Are all these being counted in the number of parameters?

**Methods And Evaluation Criteria:**

The proposed method jointly optimize for localization and editing, which is well-motivated and makes sense.

The paper also provides experiments across different model families, datasets, and model sizes

**Other Comments Or Suggestions:**

Some figures are hard to read, e.g., Figure 4 and Figure 8.

**Other Strengths And Weaknesses:**

The paper provides abundant ablations on the design choice of the method.

**Questions For Authors:**

See Experimental Designs Or Analyses.

**Relation To Broader Scientific Literature:**

The methods improves on previous methods by jointly localizing and editing the activations, which is learnt end-to-end with gating.

**Theoretical Claims:**

N/A as there aren't theoretical claims in the main paper.

---

> ### Author Rebuttal · Authors · 2025-03-31
>
> Thank you for taking the time to review our paper and provide valuable insights.
>
> Q1: Baseline setup seem either under-tuned or too weak to compare with?
>
> A1: Thank you for your feedback on our baseline system setup.
>
> (1) As noted in the original papers for each baseline method, these methods are sensitive to hyperparameters, and we detail this in Appendix D.3. The following table shows the sensitivity of these methods, with the results highlighting the lack of robustness, which motivates our proposal for a more stable approach.
> **Different learning rate in RED**
> ||SIQA|WinoGrande|Law|Physics|e2e_nlg|Web_nlg|
> |-|-|-|-|-|-|-|
> |5e-5|48.47|48.25|28.00|21.00|8.27|12.74|
> |2e-4|51.23|50.87|32.00|18.00|11.33|14.53|
> |6e-2|50.16|47.14|29.00|23.00|10.14|13.59|
> **Different prefix and suffix positions in ReFT**
> ||SIQA|WinoGrande|Law|Physics|e2e_nlg|Web_nlg|
> |p7+s7|62.51|53.96|28.00|40.00|11.44|12.57|
> |p11+s11|58.24|51.48|30.00|41.00|12.02|11.35|
> **Number of Attention Heads in LoFIT**
> ||SIQA|WinoGrande|Law|Physics|e2e_nlg|Web_nlg|
> |32|54.21|52.34|33.00|10.00|13.18|17.27|
> |64|56.52|54.56|35.00|12.00|15.56|16.34|
> |128|51.26|55.42|30.00|8.00|14.60|18.07|
>
> (2) Baseline methods require specific fine-tuning for each task, and we have 26 tasks to evaluate in our method. Fine-tuning all tasks individually is impractical due to the large hyperparameter search space. Therefore, we performed hyperparameter selection using a grid search approach. For each task, we ran a grid search with five different hyperparameter configurations, which were chosen to explore a diverse range of parameter settings that could provide the best model performance. We performed this search over key hyperparameters (e.g., learning rate, selected head/layers/positions etc. highlighted in Appendix D.3), using a validation set to select the configuration that resulted in the best performance. The results presented here correspond to the best hyperparameter configuration selected for each task based on the validation set performance. The final model was evaluated with these hyperparameters, and we averaged the results across all tasks. As observed, this had little effect on their relative performance, with our method continuing to outperform the others.
>
> Table:  best hyperparameter configuration in LLaMA
> ||Reasoning|Understanding|Generation||
> |-|-|-|-|-|
> ||ACC|ACC|BLEU|Rouge-L|
> |zero_shot|53.70|40.00|12.56|36.70|
> |BitFit|65.37|36.14|10.23|32.59|
> |RED|50.26|37.86|12.77|34.19|
> |REPE|66.04|37.43|11.49|31.04|
> |REFT|67.12|42.29|13.05|38.25|
> |LOFIT|57.74|32.71|13.14|35.51|
> |Our|70.55|47.00|17.07|40.65|
>
> (3) Given the small dataset (e.g., 200 samples in our setting), overfitting was a concern. To reduce overfitting's impact on the baseline, we used early stopping, which was not applied in the original implementation of the baseline systems. We also found that learning rate adjustment significantly affected the results. We evaluated four strategies, including linear schedule[1], Cyclic Learning Rate Schedule[2], Adaptive Heuristic Schedule[3] and Exponential Decay Schedule[4]. As shown in the following table, the exponential decay strategy proved most stable, so we used it for both the baseline and our method, as explained in Appendix D.1. The comparison of different learning rate decay strategies for JoLA and LoFIT is as follows.
> [1] Human-level control through deep reinforcement learning (Mnih et al., 2015)
> [2] Cyclical Learning Rates for Training Neural Networks (Smith et al., 2017)
> [3] A disciplined approach to neural network hyper-parameters (Smith et al., 2018)
> [4] An exponential learning rate sched- ule for deep learning (Li et al., 2019)
>
> **Different learning ratestrategies in JoLA**
> |Strategy|SIQA|WinoGrande|law|physics|e2e_nlg|web_nlg|
> |-|-|-|-|-|-|-|
> |Linear|62.71|56.49|38.00|42.00|14.05|22.83|
> |Cycle|64.25|57.26|39.00|43.00|14.37|23.44|
> |Adaptive|65.47|58.60|39.00|44.00|15.02|23.86|
> |Exponential|66.22|58.33|40.00|46.00|15.54|24.39|
> **Different learning rate strategies in LoFIT**
> |Strategy|SIQA|WinoGrande|law|physics|e2e_nlg|web_nlg|
> |Linear|54.13|53.36|35.00|6.00|13.84|16.95|
> |Cycle|54.32|54.25|34.00|6.00|14.37|17.83|
> |Adaptive|55.18|55.57|36.00|7.00|15.24|17.64|
> |Exponential|55.94|55.33|36.00|7.00|14.77|18.12|
>
> Q2: (same questions as Review QWX1 Review m3pE) parameter counting of the gating mechanism?
>
> A2: We mentioned the parameter counting in Table 3 of Appendix A, using the same calculation method as ReFT and LoFIT, where trainable parameters are divided by the parameters of the base LLM. For example, we calculated the parameters for the SIQA task using the LLaMA-3-8B model. JoLA’s parameter count considered only the interventions. Interestingly, JoLA’s count matches LoFIT’s.
>
> Q3: Some figures are hard to read, e.g., Figure 4 and Figure 8.
>
> A3: Thank you for your feedback. Due to space constraints, we combined multiple images into one, which affected readability. In the next version, we will adjust the layout to improve clarity and readability.

---

> > ### Comment · Reviewer_2Bjt · 2025-04-04
> >
> > Thank you for the clarification on the hyper-parameters. Regarding the parameter counting, I feel this way of counting might not reveal the full picture.
> >
> > I have raised my score accordingly.

---

> > > ### Author Response · Authors · 2025-04-07
> > >
> > > Thank you for the insightful feedback. We appreciate the opportunity to clarify our parameter counting in JoLA and address your concerns in detail.
> > >
> > > **1. Parameter Count Clarification**
> > > In JoLA, the trainable parameters include:
> > > + Multiplicative scaling vectors $m^{(l,i)}$ and additive bias vectors $a^{(l,i)}$ for every attention head.
> > > + HardConcrete gate parameters $\phi_{m}^{(l,i)}$ and $\phi_{a}^{(l,i)}$ for each head.
> > >
> > > During training, all these parameters are updated via gradient descent. However, the $L_0$ regularization encourages most of the gate parameters to drive their corresponding gates toward zero, effectively "pruning" the majority of the heads. At inference time, only the heads with non-zero gate expectations contribute to the model’s computation—meaning only their $m^{(l,i)}$ and $a^{(l,i)}$ are applied.
> > >
> > > **2. Comparison to LOFIT**
> > > LOFIT pre-selects a fixed subset of attention heads in a two-step process: (1) Updating parameters for all heads; (2) Fine-tuning only the selected ones.
> > > In contrast, JoLA continuously updates parameters across all heads during training. Thanks to the dynamic gating mechanism, the number of active parameters at inference is comparable to that in LOFIT.
> > > For example, consider the LLaMA-3-8B model. The training parameters can be computed as:
> > > $$P_{\text{trainable}} = \frac{D_{\text{attn}} \times (N_{\text{multi}} + N_{\text{add}} + N_{\text{gate}})}{P_{\text{LLMs}}}$$
> > > Where:
> > > + $D_{\text{attn}}$ is the dimension of each attention head,
> > > + $N_{\text{multi}}$, $N_{\text{add}}$, and $N_{\text{gate}}$ are the numbers of multiplicative, additive, and gating parameters, respectively.
> > > + $P_{\text{LLMs}}$ is the total number of parameters in the base LLM.
> > >
> > > **3. Simplified Calculation from Table 3:**
> > > + LOFIT
> > > $$\text{Trainable Parameters} = \frac{128 \times (32 \times 32 + 32 \times 32)}{8,030,257,152} + \frac{128 \times (64 + 64)}{8,030,257,152} \approx 0.00003468481 \(\text{or } 0.003468481\%)$$
> > > + JoLA
> > > $$\text{Trainable Parameters} = \frac{128 \times (32 \times 32 + 32 \times 32 + 32 \times 32 + 32 \times 32)}{8,030,257,152} \approx 0.00006528906 \ (\text{or } 0.006528906\%)$$
> > > Where, $32 \times 32$ reflects that each of the 32 layers has 32 attention heads. 64 denotes the number of selected heads in LOFIT, which can optionally be set to 128.
> > >
> > > JoLA and LOFIT maintain similar levels of activated parameters at inference. The small variations across tasks are expected: JoLA's activated heads vary dynamically by task. LOFIT's activated heads are determined by a fixed, manually set value.
> > >
> > > We plan to update Table 3 in the next version to clearly distinguish between trainable and activated parameters as follows. We believe this will provide a clearer and more comprehensive understanding of our parameter counting methodology.
> > > | Method | Total Params (%) | Active Params (%) |
> > > |--------|------------------|-------------------|
> > > | LOFIT  | 0.003468481      | 0.0002            |
> > > | JoLA   | 0.006528906      | 0.0002            |
> > >
> > > We hope this explanation addresses your concerns and helps clarify our approach.

---

### Official Review · Reviewer_QWX1 · 2025-03-15

**Overall Recommendation:** 4

**Summary:**

This paper presents a novel extension of the activation editing approach. Its primary contribution lies in integrating localization and editing into a single process using a gating mechanism, unlike previous two-stage methods that first manually locate and then edit model components. This makes the proposed method more practical and adaptive. Extensive experiments on multiple LLMs demonstrate the effectiveness of this approach, particularly in low-resource scenarios, where it outperforms existing methods.

**Claims And Evidence:**

The claims made in the submission are supported by clear and convincing evidence.

**Essential References Not Discussed:**

I did not find any essential references that were missing from the discussion.

**Experimental Designs Or Analyses:**

The soundness and validity of the experimental designs and analyses were checked.

**Methods And Evaluation Criteria:**

The proposed methods and evaluation criteria make sense for the problem and application at hand:

1. The use of a gating mechanism to jointly perform localization and editing is a reasonable approach.

2. The experiments are well-controlled by ensuring a fair comparison of trainable parameters.

**Other Comments Or Suggestions:**

I find the description of the gating mechanism not entirely clear. Specifically, the parameterized form of the Hard-Concrete distribution is not explicitly detailed. The trainable parameters of the gating mechanism are not clearly stated. It would be helpful if the authors could clarify these aspects to provide a more precise understanding of how the gating mechanism operates and how it is optimized during training.

**Other Strengths And Weaknesses:**

NA

**Questions For Authors:**

1. What exactly do the Hidden States and bias term in Fig. 2 refer to?

2. Could the reason for multiple components performing worse in Fig. 2 be overfitting? Similarly, the authors' gating mechanism demonstrates that selectively editing attention heads is more effective than editing all heads, which is somewhat counterintuitive. Could this also be due to overfitting?

3. By the end of training, how many attention heads are typically completely shut off?

4. Since the gating mechanism is modeled as a distribution, how does its randomness manifest during inference?

**Relation To Broader Scientific Literature:**

NA

**Theoretical Claims:**

NA

---

> ### Author Rebuttal · Authors · 2025-03-31
>
> Thank you for your feedback and suggestions.
> **Comment:** The parameterized form of the hard-concrete distribution is not explicitly detailed. The trainable parameters of the gating mechanism are not clearly stated.
> **Response:** The hard concrete distribution has two associated scalar parameters: a scale parameter and a temperature parameter. Following prior work on sparsification (e.g., Voita et al., 2019; Louizos et al., 2017), we train only the scale parameter and fix the temperature to 0.33.  To clarify, these gates do not take any input – each gate is simply an instance of the hard concrete distribution with a single learnable parameter.  We will clarify this in the new version of the paper.
> + **Q1: What exactly do the hidden states and bias term in Fig. 2 refer to?**
> - **A1:** Figure 2 illustrates the previously proposed forms of intervening in Transformer modules. The “Hidden states” approach follows REFT, applying interventions directly to the MLP hidden states. The “Bias” approach follows BitFit, modifying only the bias terms of attention, dropout, and layer norm activations. We will elaborate on these in Section 3.1 (currently lines 86–88) to clarify further.
> + **Q2: Could the reason for multiple components performing worse in Fig. 2 be overfitting?**
> - **A2:** We think it can be interpreted as a form of overfitting, though we do mitigate this using early stopping. More broadly, the results highlight that both deciding where we intervene (i.e., the choice and location of components) and having fewer interventions are important in our low-resource setting. As shown below, even when provided with more data (500 examples), the MLP+Attention intervention fails to match the performance of Attention-only using 200 examples across most datasets.
>
> **The same sample size as in Figure 2 (200)**
> ||SIQA|WinoGrande|Law|Physics|E2E_NLG|WEB_NLG|
> |-|-|-|-|-|-|-|
> |Attention|55.94|55.33|36.00|7.00|14.77|18.12|
> |MLP|50.10|51.62|34.00|20.00|10.31|14.45|
> **Sample size (300)**
> ||SIQA|WinoGrande|Law|Physics|E2E_NLG|WEB_NLG|
> |Attention|56.34|55.85|36.00|18.00|15.25|18.46|
> |MLP|53.07|51.93|35.00|20.00|10.68|14.91|
> **Sample size (500)**
> ||SIQA|WinoGrande|Law|Physics|E2E_NLG|WEB_NLG|
> |Attention|56.86|56.37|37.00|20.00|15.87|18.85|
> |MLP|53.92|52.45|36.00|21.00|11.22|15.26|
> **Different Sample Size (Attention+MLP)**
> |Samplesize|SIQA|WinoGrande|Law|Physics|E2E_NLG|WEB_NLG|
> |200|52.17|48.74|23.00|13.00|8.23|12.36|
> |300|52.82|49.51|25.00|14.00|8.85|12.74|
> |500|53.24|50.36|28.00|16.00|9.39|13.03|
>
> + **Q3: How many attention heads are typically completely shut off by the end of training?**
> + **A3:** To clarify, we are not shutting down any attention heads. Instead, we selectively choose which heads to apply interventions to. When both the offset gate $g_a$ and the multiplicative update gate $g_m$ are set to $0$, the head’s computation becomes identical to that of the original model. By the end of training, most gates are closed  – for example, on OBQA, 86\% of attention heads have $g_a = 0$, and 94\% have $g_m = 0$ (See Figure 8).
> + **Q4: How does the gating mechanism's randomness manifest during inference?**
> + **A4:** During training, we model each gating variable (e.g., $g_a^{(l,i)}$ and $g_m^{(l,i)}$) as random, sampled from a hard-concrete distribution. However, during inference, we use their expected values, $\mathbb{E}[g_a^{(l,i)}]$ and $\mathbb{E}[g_m^{(l,i)}]$, instead of sampling from the distributions. This removes randomness, ensuring consistency and stability in inference. We briefly mention this in lines 165-166 and will clarify in the next version.

---

> > ### Comment · Reviewer_QWX1 · 2025-04-05
> >
> > I appreciate the authors' response and the additional experiments, which have addressed all of my concerns. Therefore, I have decided to raise my score. I hope the authors will include these clarifications from the rebuttal in the next version of the paper, as I believe they are important for understanding your work.

---

> > > ### Author Response · Authors · 2025-04-07
> > >
> > > Thank you for your valuable feedback and for raising the score. We are glad the additional experiments addressed your concerns. We will include the clarifications from the rebuttal in the next version of the paper for better clarity.

---

### Decision · Program_Chairs · 2025-05-01

**Decision:**

Accept (poster)

**Comment:**

This paper presents JoLA (Joint Localization and Activation Editing), a novel and practical method for parameter-efficient fine-tuning (PEFT) of large language models (LLMs) in low-resource settings. JoLA introduces a unified framework that simultaneously: 1). Identifies which attention heads to intervene on, 2). Learns the type of intervention (additive, multiplicative, or both), 3). Optimizes the associated intervention parameters. A core innovation lies in its use of Hard Concrete gating with expected-L0 regularization, allowing selective and sparse updates.

JoLA is a well-motivated, rigorously evaluated, and practically impactful method for PEFT in LLMs. The approach is both novel in design and strong in empirical performance [JoLA consistently outperforms baselines across 26 tasks and multiple model families, including LLaMA, Qwen, and Mistral, under low-resource settings (e.g., ~200 samples)], with substantial improvements over state-of-the-art baselines in low-resource settings. I recommend acceptance.